# Membrane estrogen receptor alpha (ERα) participates in flow-mediated dilation in a ligand-independent manner

Julie Favre[1†], Emilie Vessieres[1,2†], Anne-Laure Guihot[1,2], Coralyne Proux[1,2], Linda Grimaud[1], Jordan Rivron[1,2], Manuela CL Garcia[1,2], Léa Réthoré[1], Rana Zahreddine[3], Morgane Davezac[3], Chanaelle Fébrissy[3], Marine Adlanmerini[3], Laurent Loufrani[1,4], Vincent Procaccio[1,4], Jean-Michel Foidart[5], Gilles Flouriot[6], Françoise Lenfant[3], Coralie Fontaine[3], Jean-François Arnal[3], Daniel Henrion[1,2,4]*

[1]Angers University, MITOVASC, CNRS UMR 6015, INSERM U1083, Angers, France; [2]CARFI facility, Angers University, Angers, France; [3]INSERM U1297, Paul Sabatier University (Toulouse III) , University Hospital (UHC) of Toulouse, Toulouse, France; [4]University Hospital (CHU) of Angers, Angers, France; [5]Groupe Interdisciplinaire de Génoprotéomique Appliquée, Université de Liège, Liège, Belgium; [6]INSERM U1085, IRSET (Institut de Recherche en Santé, Environnement et Travail), University of Rennes, Rennes, France

**Abstract:** Estrogen receptor alpha (ERα) activation by estrogens prevents atheroma through its nuclear action, whereas plasma membrane-located ERα accelerates endothelial healing. The genetic deficiency of ERα was associated with a reduction in flow-mediated dilation (FMD) in one man. Here, we evaluated ex vivo the role of ERα on FMD of resistance arteries. FMD, but not agonist (acetylcholine, insulin)-mediated dilation, was reduced in male and female mice lacking ERα (*Esr1-/-* mice) compared to wild-type mice and was not dependent on the presence of estrogens. In C451A-ERα mice lacking membrane ERα, not in mice lacking AF2-dependent nuclear ERα actions, FMD was reduced, and restored by antioxidant treatments. Compared to wild-type mice, isolated perfused kidneys of C451A-ERα mice revealed a decreased flow-mediated nitrate production and an increased $H_2O_2$ production. Thus, endothelial membrane ERα promotes NO bioavailability through inhibition of oxidative stress and thereby participates in FMD in a ligand-independent manner.

**\*For correspondence:**
daniel.henrion@univ-angers.fr

†These authors contributed equally to this work

**Competing interest:** The authors declare that no competing interests exist.

## Editor's evaluation

Using multiple genetically modified mouse models, the authors have demonstrated a novel role of membrane associated estrogen receptor alpha (ERα) signaling to modulate flow-mediated dilation (FMD) in a ligand-independent manner. Specifically, the results indicate that non-nuclear actions of membrane estrogen receptor α in endothelial cells support flow-mediated vasodilation in animals of both sexes via mechanisms that are independent of estrogenic ligands, involving NO production and an attenuation of the NO-inactivating effects of reactive oxygen species. These findings highlight a novel role of ligand-independent activation of membrane estrogen receptor α in regulation of vascular physiology and possibly in disease, adding to the recently introduced paradigm shift in the understanding of estrogen and estrogen receptor function.

## Introduction

Resistance arteries are the small blood vessels located upstream of capillaries. Alteration of their structures or functions can raise capillary pressure, which exacerbates organ damage due to cardio- and cerebro-vascular risk factors and associated organ disorders. The basal tone of resistance arteries allows for tight control of local blood flow. This tone results from the interaction between pressure-induced smooth muscle contraction and flow-mediated dilation (FMD) due to the activation of endothelial cells by shear stress. FMD measured in the human forearm depends mainly on the acute production of NO by endothelial cells in response to an acute increase in shear stress (*Joannides et al., 1995*; *Green et al., 2014*; *Zhou et al., 2014*) and reduced FMD is a hallmark of endothelium dysfunction (*Zhou et al., 2014*; *Rizzoni and Agabiti Rosei, 2006*; *Stoner and Sabatier, 2012*).

Epidemiological investigations have shown that, prior to menopause, women are less affected by cardiovascular disorders than men (*Simoncini, 2009*; *Arnal et al., 2017*). Estrogens protect against atherosclerosis (*Billon-Galés et al., 2009*) and neointimal proliferation (*Smirnova et al., 2015*), and accelerate re-endothelialization of injured arteries (*Brouchet et al., 2001*). Numerous actions of 17-beta-estradiol (E2) are mediated by estrogen receptor alpha (ERα), which acts in the nucleus as a transcription factor. E2 is strongly involved in the outward remodelling of the uterine blood vessels during pregnancy (*Mandala and Osol, 2012*). Indeed, we have previously shown that E2 and ERα, and more precisely its nuclear activating function AF2, are both essential for the arterial outward remodeling induced by a chronic rise in blood flow in vivo (*Tarhouni et al., 2013*; *Tarhouni et al., 2014a*; *Tarhouni et al., 2014b*).

However, a subpopulation of ERα is also associated with the plasma membrane and activates non-nuclear signaling (*Arnal et al., 2017*; *Banerjee et al., 2014*; *Lu et al., 2017*). The acute effect initially described in 1967 was a rapid increase in AMPc production in the rat uterus in response to E2 (*Szego and Davis, 1967*). E2 binding to the plasma membrane was subsequently reported in endometrial cells and hepatocytes (*Pietras and Szego, 1977*), suggesting that a fraction of ERα could be located to the membrane and contributes to the rapid effects of E2, possibly through the rapid activation of G proteins and kinases such as ERK1-2, PI3K, or P21ras (*Arnal et al., 2017*). In ovine fetal pulmonary artery endothelial cells, E2 stimulates eNOS activity through activation of ERα leading to increased intracellular $Ca^{2+}$ within minutes (*Lantin-Hermoso et al., 1997*). By contrast, in HUVECs, E2 induces a rapid production of NO and cGMP independent of an increase in intracellular $Ca^{2+}$ (*Caulin-Glaser et al., 1997*). This rapid nongenomic activation of eNOS involves Akt/PKB (*Florian et al., 2004*) and MAP kinase-dependent mechanisms (*Chen et al., 1999*). Estradiol-induced endothelium-independent dilation was also described in canine coronary arteries (*Sudhir et al., 1995*) and in rat cerebral microvessels (*Florian et al., 2004*). This dilation is also mediated by ERα located at the level of the plasma membrane. Using a mouse model lacking membrane-associated ERα, we demonstrated that the acute vasodilator effect of E2 and its accelerative effect on re-endothelialization are mediated by membrane-associated ERα (*Adlanmerini et al., 2014*; *Zahreddine et al., 2021*). On the other hand, E2 exerts protective effects against atheroma, angiotensin 2-induced hypertension, and neointimal hyperplasia through its nuclear effects (*Guivarc'h et al., 2018*).

The 7-transmembrane G-protein-coupled estrogen receptor (GPER, formerly known as GPR30) is another receptor located not only at the plasma membrane but also on the membrane of the endoplasmic reticulum that can be activated by E2. It was found in both human and animal arteries (*Prossnitz and Barton, 2011*; *Barton et al., 2018*). The combination of GPER-selective agonists and antagonists with the use of GPER-knock-out mice allowed to elucidate more specifically its biological effects arteries (*Prossnitz and Barton, 2011*; *Barton et al., 2018*). In the rat, GPER activation by its agonist G-1 reduces uterine vascular tone during pregnancy through activation of endothelium-dependent NO production (*Tropea et al., 2015*). Likewise, the G-1-induced relaxation of the mesenteric resistance arteries in both male and female rats is mainly mediated by the PI3K-Akt-eNOS pathway (*Peixoto et al., 2017*). Noteworthy, GPER partly contributes to E2-dependent vasodilation of mouse aortae (*Fredette et al., 2018*). Thus, both ERα and GPER could contribute to the rapid actions of E2, although their respective roles according to vessel type, species and pathophysiological context remain to be established.

The risk of cardiovascular diseases differs between men and women, and the protection of women is progressively lost after menopause. For instance, endothelium-dependent dilation of subcutaneous arteries is reduced in post-menopausal women compared to pre-menopausal women (*Kublickiene*

*et al., 2005*; *Kublickiene et al., 2008*). This protection involves NO production in response to estrogens. Similarly, diet phytoestrogens could have protective actions in postmenopausal women suffering coronary artery disease (*Cruz et al., 2008*) and selective estrogen receptor modulators (SERMs) such as raloxifene exert protective actions in female rats through eNOS activation (*Chan et al., 2010*). Besides the activation of eNOS, hormonal replacement therapy also activates endothelium-derived hyperpolarizing factor (EDHF)-mediated vasodilation as shown in rat mesenteric and uterine arteries (*Burger et al., 2009*) as well as in the rat gracilis muscle artery with increased Epoxyeicosatrienoic acids (EETs) production involved in E2-mediated increase in FMD in hypertensive or old rats (*Huang et al., 2001*; *Sun et al., 2004*). Furthermore, estrogen therapy reduces pressure (myogenic) (*Kublickiene et al., 2005*; *Kublickiene et al., 2008*) and adrenergic-dependent contraction (*Meyer et al., 1997*). FMD is also improved by E2 in rat gracilis muscle arteries (*Huang et al., 1998*). Although there is an increase in the amplitude of FMD in women among the menstrual cycle with a greater dilation during the luteal or follicular phase (*Hashimoto et al., 1995*), FMD is similar in healthy young men and women (*Sullivan et al., 2015*). Importantly, the first disruptive mutation in the gene encoding ERα, reported in 1994 in a man who was only 30 years old (*Smith et al., 1994*), was found to be associated with a total absence of FMD (*Sudhir et al., 1997*). This single yet major clinical observation suggests that ERα-dependent signal transduction could play a role in FMD in males. Of note, conversion of testosterone into estradiol by aromatase which is expressed in the arterial wall has been shown to reduce early atherogenesis in male mice (*Nathan et al., 2001*).

In the present study, we investigated the role of ERα and its different subfunctions on FMD in isolated mouse resistance arteries. To this aim, we used different mouse models that were: (i) fully deficient in ERα (*Esr1*[-/-] mice), (ii) deleted in seven amino acid in the helix 12 and thus deficient in activation function (AF)–2 necessary for the nuclear transcriptional activity of ERα (AF2[0]ERα mice), and (iii) invalidated for plasma membrane-associated signaling. To explore the role of membrane ERα, we used: (1) mice that carry a mutation of the codon encoding the cysteine (Cys) 451 palmitoylation site of ERα to alanine (C451A-ERα mice) so that the anchoring of the receptor to the plasma membrane is impossible (*Adlanmerini et al., 2014*) and (2) a knock-in mouse model of ERα mutated for the arginine 264 (R264A-ERα mice) so that the interaction of the membrane-located ERα with Gαi involved in rapid NO production is suppressed (*Adlanmerini et al., 2020*). Moreover, because the absence of membrane-associated ERα strongly reduced FMD, additional experiments were conducted to investigate: (i) the involved mechanisms, (ii) its counterpart in female mice, and (iii) the potential role of ERα activation by its ligands.

## Results

### FMD is reduced in mice lacking ERα but unaffected by exogenous estrogens

In mice lacking ERα (see the scheme in *Figure 1A*), FMD was significantly reduced in resistance arteries isolated from male *Esr1*[-/-] mice compared to littermate *Esr1*[+/+] mice (*Figure 1B*). Precontraction prior to FMD (*Figure 1C*) and arterial diameter (*Figure 1D*) were not significantly affected by the absence of ERα. Agonist-mediated endothelium-dependent (acetylcholine and insulin) and endothelium-independent (SNP) dilation were not significantly affected by the absence of ERα (*Figure 1E–G*).

To directly investigate the influence of estrogens on FMD, mesenteric resistance arteries isolated from male WT mice were incubated (20 min) with E2, which activates both membrane-associated and nuclear ERα, or with another natural estrogen, estetrol (E4), which activates only nuclear ERα (*Abot et al., 2014*). These exogenous estrogens did not affect FMD (*Figure 1H*). Furthermore, the estrogen receptor downregulator and GPER agonist fulvestrant (ICI-182780) (*Meyer et al., 2010*; *Jacenik et al., 2016*) did not alter FMD after 20 min of incubation with isolated arteries (*Figure 1H*). Similarly, the GPER antagonist G-36 did not alter FMD (*Figure 1I*) although G-36 inhibited the dilation induced by both E2 and the GPER agonist G-1 (*Figure 1J*).

We also found that FMD in mesenteric resistance arteries was similarly reduced in both intact (*Figure 1K*) and ovariectomized female *Esr1*[-/-] mice (*Figure 1L*) compared to their respective *Esr1*[+/+] littermate controls. The similar levels of FMD in intact and ovariectomized female WT mice, as well as in male mice, suggest that circulating endogenous estrogens do not influence FMD in young healthy

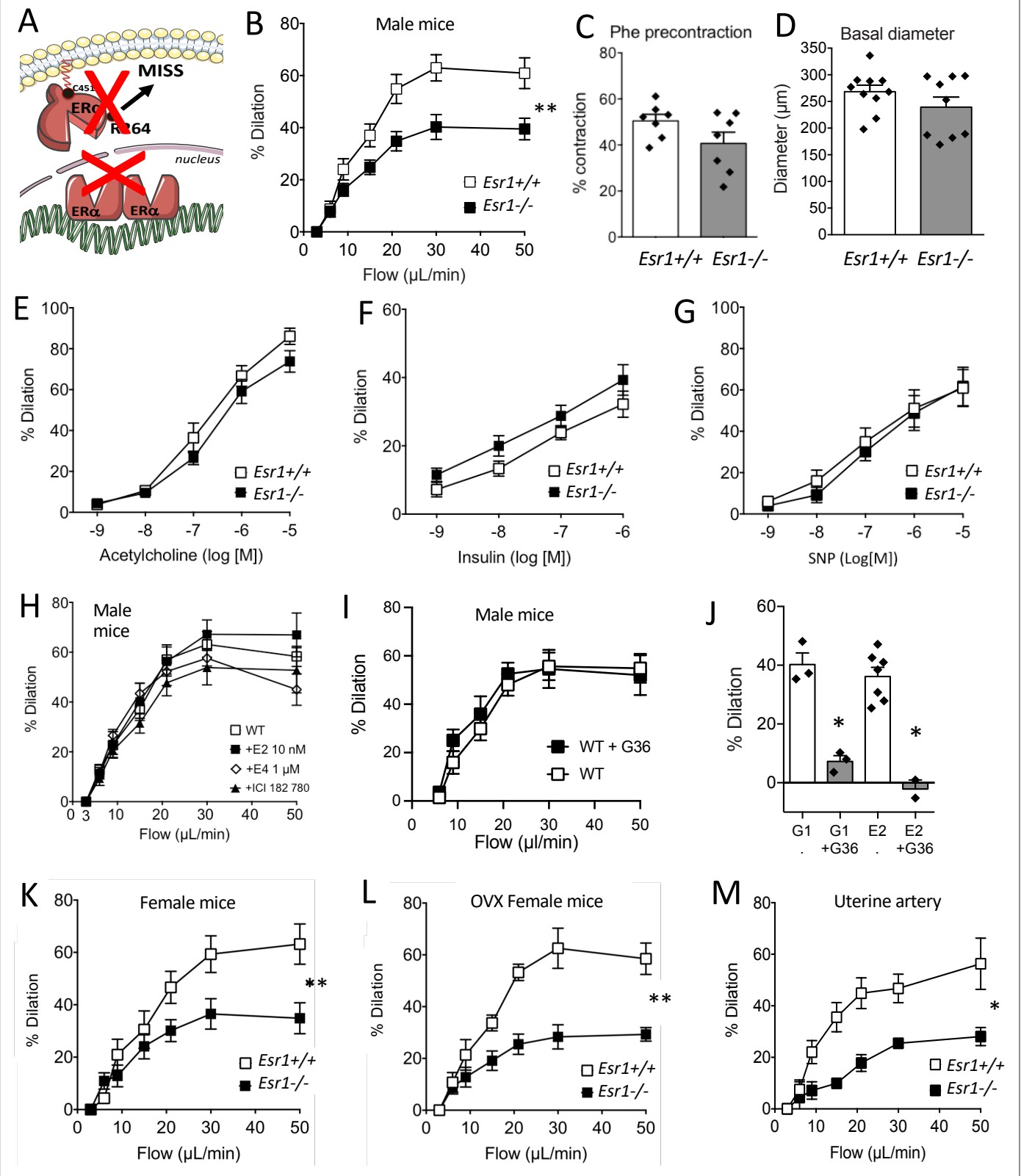

**Figure 1.** Involvement of ERα in flow-mediated dilation (FMD). FMD was measured in mesenteric resistance arteries isolated from male mice lacking ERα (*Esr1⁻/⁻*) and male wild-type littermates (*Esr1⁺/⁺*) (**A**). (**B**) FMD was determined in response to stepwise increases in luminal flow in male *Esr1⁻/⁻* and *Esr1⁺/⁺* mice. (**C**) Precontraction with phenylephrine (Phe) before measurement of FMD. (**D**) Basal diameter of the arteries used for FMD measurment. Besides FMD, acetylcholine- (**E**), insulin- (**F**), and sodium nitroprusside- (SNP, **G**) mediated dilation was measured in mesenteric resistance arteries

*Figure 1 continued on next page*

*Figure 1 continued*

isolated from male *Esr1*[-/-] and *Esr1*[+/+] mice. FMD was also measured in wild-type (WT) mice in the presence (20 min incubation) or absence of 17-β-estradiol (E2, 0.01 μmol/L, **H**), estetrol (E4, 1 μmol/L, **H**), ICI 182 780 (1 μmol/L, **H**) and the GPER antagonist G-36 (10 μM, **I**). (**I**) G-1 (10 μM)- and E2 (0.01 μM)-mediated dilation in the presence or absence of G-36 (1 μM). FMD was then measured in mesenteric arteries isolated from intact (**K**) and ovariectomized (OVX, **L**) female *Esr1*[-/-] and *Esr1*[+/+] mice as well as in and uterine arteries from *Esr1*[-/-] and *Esr1*[+/+] mice (**M**). Flow rate rate was 3, 6, 9, 12, 15, 30, and 50 μl/min corresponding to 0.8, 1.2, 2, 2.8, 4, 8, and 12 dyn/cm². Means ± the SEM are shown (n = 7–18 mice per group). Two-way ANOVA for repeated measurements: p = 0.0072 (interaction: p < 0.0001, **B**), p = 0.0087 (interaction: p < 0.0001, **K**), p = 0.0030 (interaction: p < 0.0001, **L**), p = 0.0119 (interaction: 0.0107, **M**). NS: two-way ANOVA for repeated measurements, panel E to I. NS: Two-tailed Mann-Whitney test, panels C and D. See source data in *Figure 1—source data 1*.

The online version of this article includes the following figure supplement(s) for figure 1:

**Source data 1.** Data and statistical analysis from experiments plotted in *Figure 1B—M*.

mice. FMD was also reduced in the uterine artery isolated from female *Esr1*[-/-] mice in comparison with *Esr1*[+/+] mice (*Figure 1M*).

FMD was not altered by the inactivation of *Esr2*, encoding ERβ in mice (*Figure 2A*). Similarly, arterial precontraction, basal diameter and acetylcholine-mediated dilation were not significantly affected by the absence of ERß (*Figure 2B–D*).

As FMD depends on the response of the endothelium to shear stress, we next investigated FMD in mice lacking ERα in endothelial cells (*Tek*[Cre/+]: *Esr1*[f/f] mice). FMD was reduced in arteries isolated from these mice compared to littermate WT mice (*Figure 2E*). Arterial precontraction, basal diameter

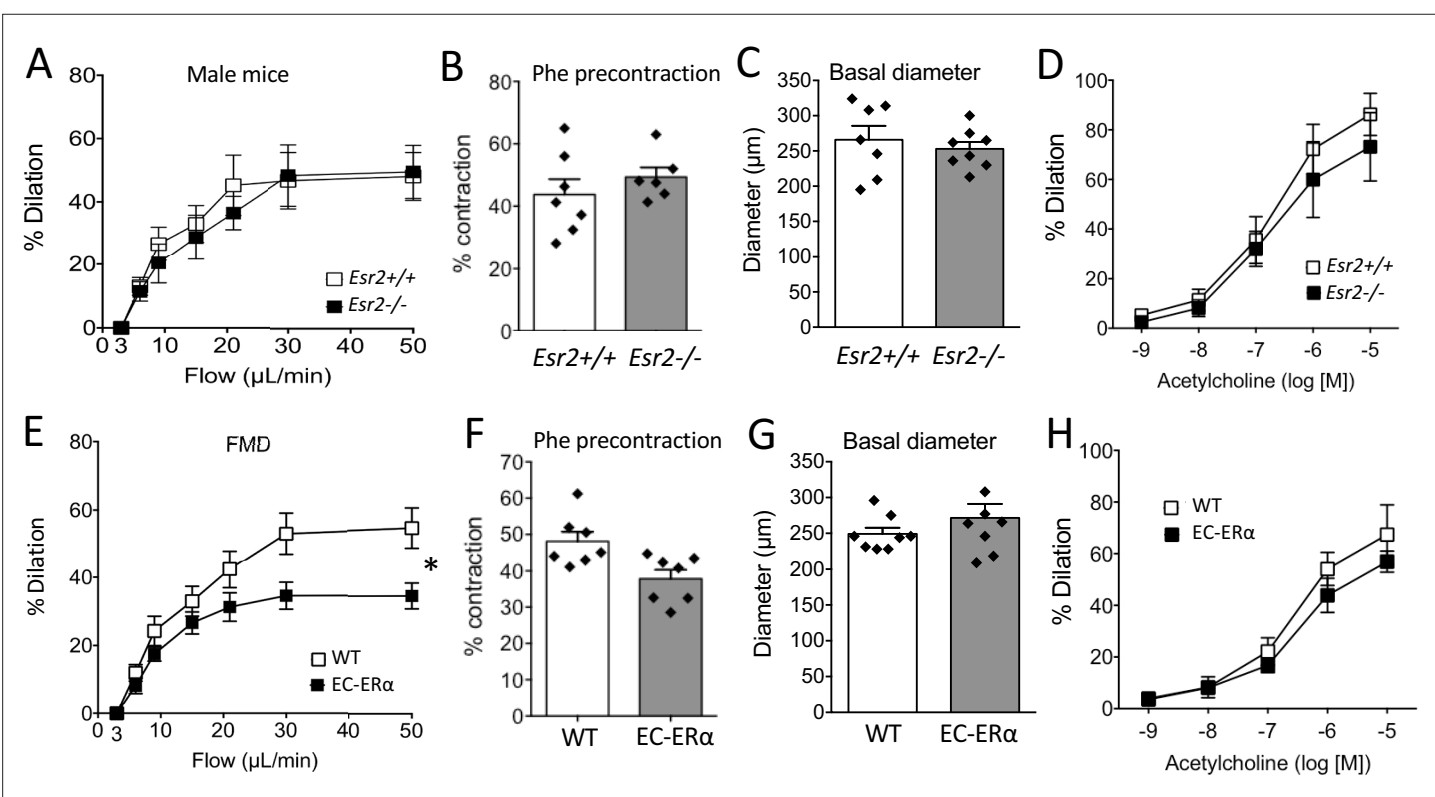

**Figure 2.** Involvement of ERβ and endothelial ERα in flow-mediated dilation (FMD). (**A to D**) FMD, precontraction, basal diameter and acetylcholine-mediated dilation measured in male mice lacking ERβ (*Esr2*[-/-]) and their littermate control (*Esr2*[+/+]). (**E to H**) FMD, precontraction, basal diameter and acetylcholine-mediated dilation measured in Tek[Cre/+]:*Esr1*[-/-] male mice lacking endothelial ERα (EC-ERα) and Tek[Cre/-]:*Esr1*[lox/lox] their littermate controls (WT). Flow rate rate was 3, 6, 9, 12, 15, 30, and 50 μl/min corresponding to 0.8, 1.2, 2, 2.8, 4, 8, and 12 dyn/cm².+ source data 2. Means ± the SEM are shown (n = 6 or 7 mice per group). Two-way ANOVA for repeated measurements: p = 0.0273 (interaction: p = 0.0069**, **E**). NS: two-way ANOVA for repeated measurements, panel A, D, and H. NS: two-tailed Mann-Whitney test, B, C, F, and G. Data and analysis in *Figure 2—source data 1*.

The online version of this article includes the following figure supplement(s) for figure 2:

**Source data 1.** Data and statistical analysis from experiments plotted in *Figure 2A—H*.

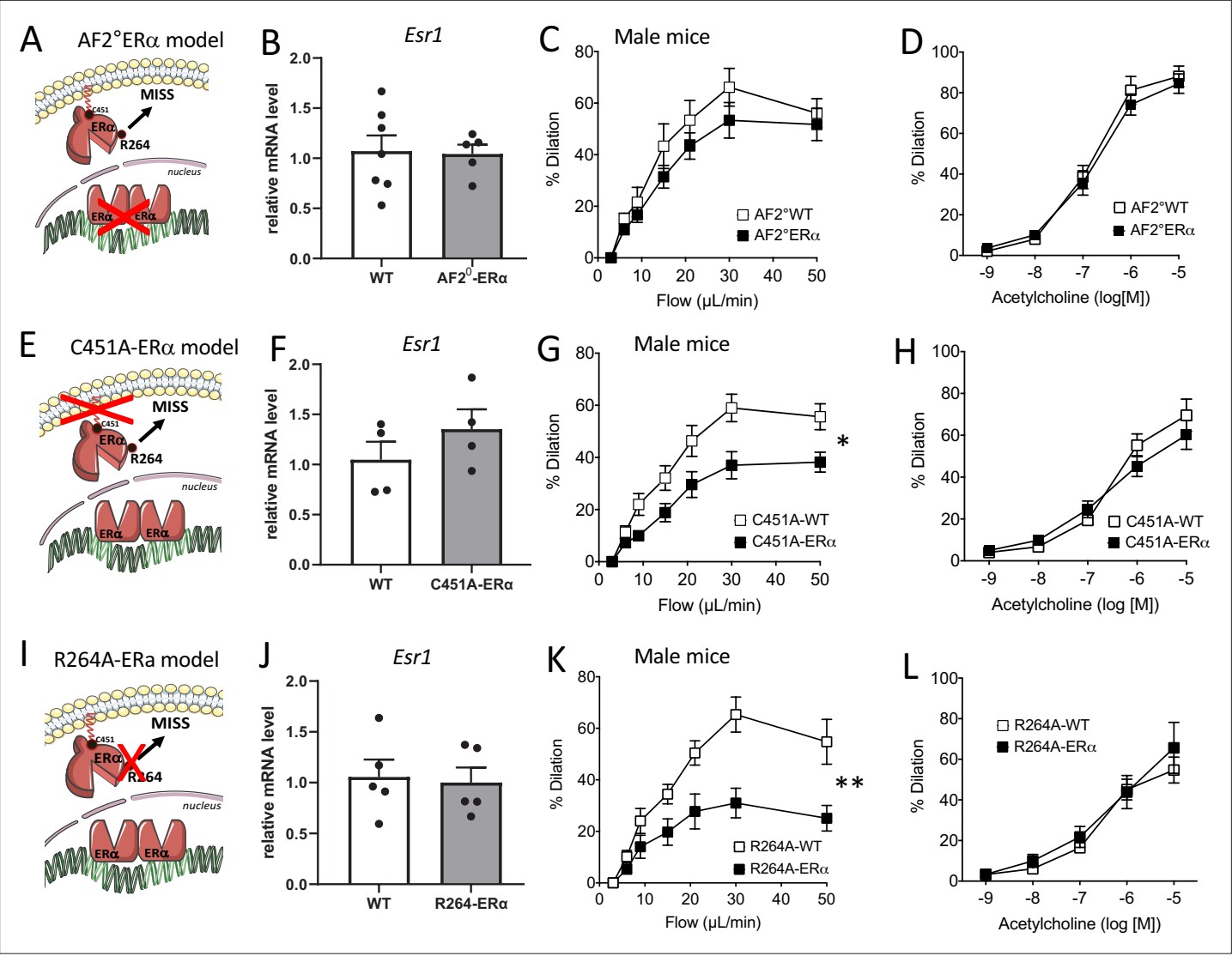

**Figure 3.** Flow-mediated dilation in mice lacking nuclear or membrane-associated ERα. *Esr1* expression level in aortic endothelial cells (expression relative to the housekeeping genes *Gapdh, Hprt* and *Gusb*), flow-mediated dilation (FMD) acetylcholine-mediated dilation were measured in mesenteric resistance arteries isolated from AF2-WT and AF2$^0$ERα male mice (**A to D**), C451A-WT and C451A-ERα male mice (**E to H**) and R264A-WT and R264A-ERα male mice (**I to L**). Means ± the SEM is shown (n = 13 AF2$^0$ERα, n = 5 AF2-WT mice, n = 8 C451A-ERα, n = 6 C451A-WT mice, n = 9 R264A-WT and n = 10 R264A-ERα mice). Flow rate rate was 3, 6, 9, 12, 15, 30, and 50 µl/min corresponding to 0.8, 1.2, 2, 2.8, 4, 8, and 12 dyn/cm². Two-way ANOVA for repeated measurements: panel **C**, p = 0.2681 (interaction: p = 07302), panel **G**, p = 0.0114 (interaction: p = 0.002), panel **K**, p = 0.0015 (interaction: p = 0.0002). Panels **D**, **H**, and **L**: NS. NS, two-tailed Mann-Whitney test (panels B, **F and J**).

The online version of this article includes the following figure supplement(s) for figure 3:

**Figure supplement 1.** Markers of endothelial and smooth muscle cells in mouse aortic endothelial cells.

**Figure supplement 1—source data 1.** Data and statistical analysis from experiments plotted in *Figure 3B–D,F–H, and J–L* and for *Figure 3—figure supplement 1A–F*.

and acetylcholine-mediated dilation were not significantly affected by the absence of endothelial ERα (*Figure 2F–H*).

Altogether, these results demonstrate a crucial role of ERα in FMD in both males and females and probably in a ligand independent manner. We, therefore, decided to use male mice for the remainder of the study.

## FMD in mice lacking the nuclear activation function 2 (AF2) of ERα

As the AF2 nuclear function mediates several protective effects of ERα on the vasculature (*Guivarc'h et al., 2018*), we first investigated FMD in AF2⁰ERα mice (*Figure 3A*). The gene expression level of *Esr1* (encoding ERα) in endothelial cells was not affected by invalidation of the AF2 function of ERα (*Figure 3B*). Quality of mRNA endothelial enrichissment was attested using analysis of *Tek* expression as a marker of endothelial cells and *Cnn1* expression as a marker of smooth muscle cells (*Figure 3—figure supplement 1*).

We observed that FMD was not significantly reduced in mesenteric resistance arteries isolated from AF2⁰ERα male mice compared to littermate AF2⁰WT animals (*Figure 3C*). Acetylcholine-mediated dilation (*Figure 3D*) was not altered by loss of the nuclear AF2 function of ERα. Thus, we demonstrated that FMD was preserved despite the loss of AF2 nuclear function of ERα.

## FMD in mice lacking membrane-located ERα effects

We thus evaluated the role of the membrane ERα in FMD, thanks to two complementary models, C451A-ERα mice[24] and R264A-ERα mice (*Adlanmerini et al., 2020*), that allowed us to previously investigate the ERα membrane-initiated steroid signaling (MISS) pathway in the accelerative effect of E2 on re-endothelialization following arterial injury and in the acute dilation induced by E2 through rapid eNOS activation.

First, we investigated FMD in C451A-ERα male mice that lack the capacity to anchor ERα to the plasma membrane through palmitoylation (*Adlanmerini et al., 2014*; *Figure 3E*). The gene expression level of *Esr1* was not significantly altered in aortic endothelial cells isolated from C451A-ERα mice compared to wild-type mice (*Figure 3F*). FMD was significantly reduced compared to WT littermate animals (*Figure 3G*) while acetylcholine-mediated dilation was not significantly affected by the absence of membrane-associated ERα (*Figure 3H*).

We then investigated FMD in R264A-ERα male mice that lack the capacity to activate the $G_{\alpha i}$ involved in acute NO production upon membrane-ERα activation (*Figure 3I*; *Adlanmerini et al., 2020*). The gene expression level of *Esr1* was not significantly altered in aortic endothelial cells isolated from R264A-ERα mice compared to wild-type mice (*Figure 3J*). We found that FMD was significantly reduced in R264A-ERα male mice compared to WT littermate controls (*Figure 3K*). On the other hand, acetylcholine-mediated dilation was not significantly affected by the absence of membrane-associated ERα effects in R264A-ERα male mice (*Figure 3L*).

Thus, we demonstrated that FMD was altered as a consequence of either the inactivation of palmitoylation site of ERα (C451A-ERα mice) or the impairment of the activation of the $G_{\alpha i}$ protein by ERα (R264A-ERα mice), thus preventing membrane-associated ERα activation of FMD in male mice.

## Role of ERα in the activation of the NO pathway in FMD

As NO produced by endothelial NOS plays a key role in endothelium-dependent dilation and thus in FMD, we investigated the effect of the inhibition of NO synthesis by L-NNA on FMD and acetylcholine-mediated dilation. FMD was significantly reduced by L-NNA in mesenteric resistance arteries of the four groups of littermate WT mice (*Figure 4A–D*), whereas L-NNA had no significant effect on FMD in *Esr1*⁻/⁻ (*Figure 4A*), C451A-ERα (*Figure 4B*), and R264A-ERα mice (*Figure 4C*). In contrast, FMD was reduced to a similar extent by L-NNA in AF2⁰ERα and WT mice (*Figure 4D*).

L-NNA strongly and similarly reduced acetylcholine-mediated relaxation in *Esr1*⁻/⁻ (*Figure 4E*), C451A-ERα (*Figure 4F*), R264A-ERα (*Figure 4G*), AF2⁰ERα (*Figure 4H*) and the corresponding littermate WT mice (*Figure 4E–H*), thus showing that the alteration of membrane ERα activation affected selectively the flow-mediated NO-dependent dilation, but not the acetylcholine-mediated NO-dependent dilation. The addition of either the inhibitor of cyclooxygenase indomethacin (*Figure 4A–H*) or the inhibitor epoxyeicosatrienoic acids (EETs) synthesis MSPPOH (*Figure 4—figure supplement 1A–F*) did not further reduced FMD or acetylcholine-mediated relaxation in all the groups.

Western-blot analysis of eNOS and phosphorylated eNOS was then performed on isolated resistance arteries that had been mounted in an arteriograph and then submitted or not to flow during 2 min (*Figure 5*). This flow rate is equivalent to the maximal response to flow observed in arteriography. In WT mice, the phosphorylation of eNOS at Ser1177 by flow (shear stress) was greater in arteries submitted to flow than in control (no flow) arteries as evidenced by a greater ratio of phosphorylated eNOS/total eNOS (*Figure 5A*). This ratio was not significantly greater in arteries submitted

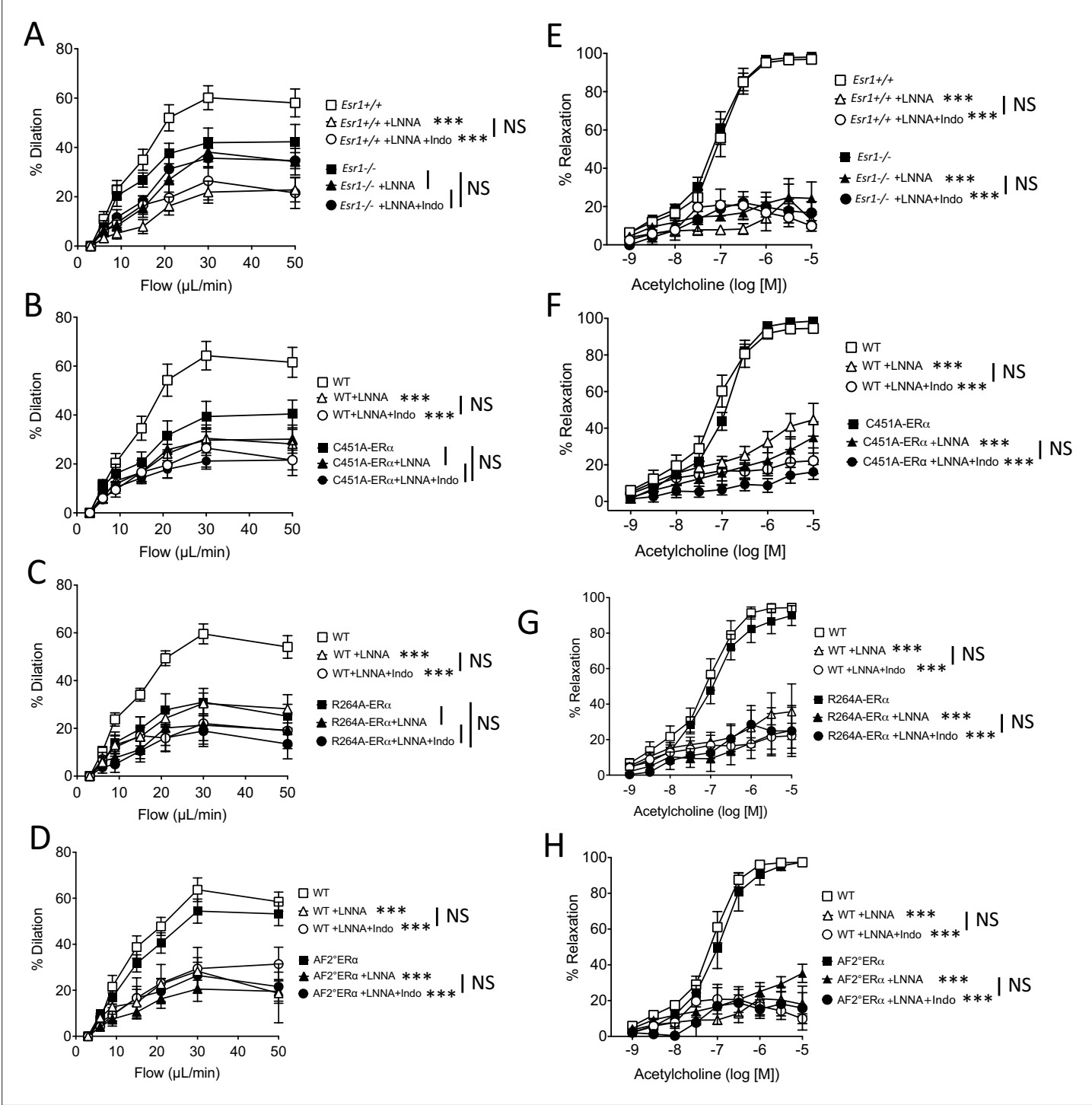

**Figure 4.** with one supplement: Effect of the blockade of NO synthesis and cyloxygenase on flow-mediated dilation. Flow-mediated dilation (FMD) was determined in pressurized mesenteric resistance arteries isolated from male *Esr1$^{+/+}$* and *Esr1$^{-/-}$* (**A**), C451A-WT and C451A-ERα (**B**), R264A-WT and R264A-ERα (**C**), AF2$^0$WT and AF2$^0$ERα mice (**D**), before and after addition of the NO synthesis blocker L-NNA (100 μM, 30 min) and then of the combination of L-NNA plus indomethacin (indo, 10 μM, 30 min). Acetylcholine-mediated relaxation was measured in the same groups in the presence and in the absence of L-NNA and of L-NNA plus indomethacin (**E to H**). Flow rate rate was 3, 6, 9, 12, 15, 30, and 50 μl/min corresponding to 0.8, 1.2, 2, 2.8, 4, 8, and 12 dyn/cm$^2$. Means ± the SEM are shown (n = 6–8 per group). ***p < 0.001, two-way ANOVA for repeated measurements, L-NNA or L-NNA+ indo versus untreated arteries within each group. Data and analysis in *Figure 4—source data 1*.

The online version of this article includes the following figure supplement(s) for figure 4:

*Figure 4 continued on next page*

*Figure 4 continued*

**Source data 1.** Data and statistical analysis from experiments plotted in *Figure 4A–H*.

**Figure supplement 1.** Effect of the blockade of NO synthesis (L-NNA), cyclooxygenase (indomethacin) and EETs production (MSPPOH) on flow-mediated dilation.

**Figure supplement 1—source data 1.** Data and statistical analysis from experiments plotted in *Figure 4—figure supplement 1A–G*.

to flow than in unstimulated arteries in C451A-ERα mice (*Figure 5A*). The expression level of total eNOS was similar in C451A-ERα and in WT mice (*Figure 5B*). A similar pattern was observed in R264A-ERα mice (*Figure 5D and E*). By contrast, the ratio of phosphorylated eNOS/total eNOS was similarly increased by flow in WT and AF2-ERα mice without any change in total eNOS level between the two strains (*Figure 5G and H*).

Western-blot analysis of Akt and phosphorylated Akt was then performed on the same samples and a similar pattern was observed (*Figure 5C,F,I*). All the blots are shown in *Figure 5—source data 2*.

These results show that the absence of membrane-associated ERα affects flow-mediated eNOS activation pathway, at least in part, by preventing the activation of its upstream activator Akt/PKB.

## Flow-mediated dilation and $NO_2/NO_3$ production in the isolated perfused kidney

As the kidney is a well-known autoregulated organ with a dense microvascular network, we investigated flow-mediated responsiveness in perfused kidneys isolated from C451A-ERα and WT mice (*Figure 6A*). First, the flow-pressure relationship was shifted leftward in C451A-ERα mice compared to WT mice (*Figure 6B*), suggesting reduced endothelial responsiveness to flow. Acetylcholine-mediated dilation in perfused kidneys was equivalent in C451A-ERα and WT mice (*Figure 6C*), suggesting that the response to flow was probably selectively reduced in perfused kidney of C451A-ERα mice, as shown above for the mesenteric artery. Similarly, phenylephrine-mediated contraction was not affected by the absence of membrane-ERα (51.1 ± 2.9 vs 57.4% ± 5.7% contraction, C451A-ERα and WT mice, n = 5 per group, p > 0.9999, Mann-Withney test).

Then, we measured nitrate and nitrite concentration in the kidney perfusate and found that it was reduced in C451A-ERα compared to WT mice (*Figure 6D*). Interestingly, ATP production measured in the kidney perfusate was also reduced in C451A-ERα mice (*Figure 6E*) whereas $H_2O_2$ production was higher in C451A-ERα than in WT mice (*Figure 6F*).

Thus, these results suggest that FMD reduction due to the absence of membrane-ERα also affects the capacity of the renal vasculature to produce NO and ATP, whereas the increased $H_2O_2$ production suggests an excessive oxidative stress in response to flow in C451A-ERα mice.

## Loss of membrane-associated ERα did not affect gene expression

Finally, we analyzed in mesenteric resistance arteries the expression of 44 genes that may be involved in the rapid endothelial response to acute changes in flow (FMD). No significant difference was observed between C451A-ERα and WT mice (*Figure 7—figure supplements 1 and 2*), in line with the prominent or even exclusive role of rapid, non genomic, membrane-ERα.

## Acute pharmacological ROS reduction restored FMD in C451A-ERα mice

As the production of $H_2O_2$ in the kidneys from C451A-ERα mice was higher than in WT mice, we measured FMD in arteries from C451A-ERα and WT mice after pretreatment with various antioxidants. First, the addition of PEG-SOD plus catalase to the bath containing mesenteric resistance arteries isolated from WT mice did not alter FMD (*Figure 7A*). By contrast, PEG-SOD plus catalase enhanced FMD in arteries from C451A-ERα mice (*Figure 7B*).

Catalase alone did not alter FMD in arteries from WT mice (*Figure 7C*), whereas it reduced FMD in arteries from C451A-ERα mice (*Figure 7D*).

Inhibition of mitochondrial ROS production by Mito-Tempo did not affect FMD in arteries from WT mice (*Figure 7E*), while it enhanced FMD in arteries isolated from C451A-ERα mice (*Figure 7F*).

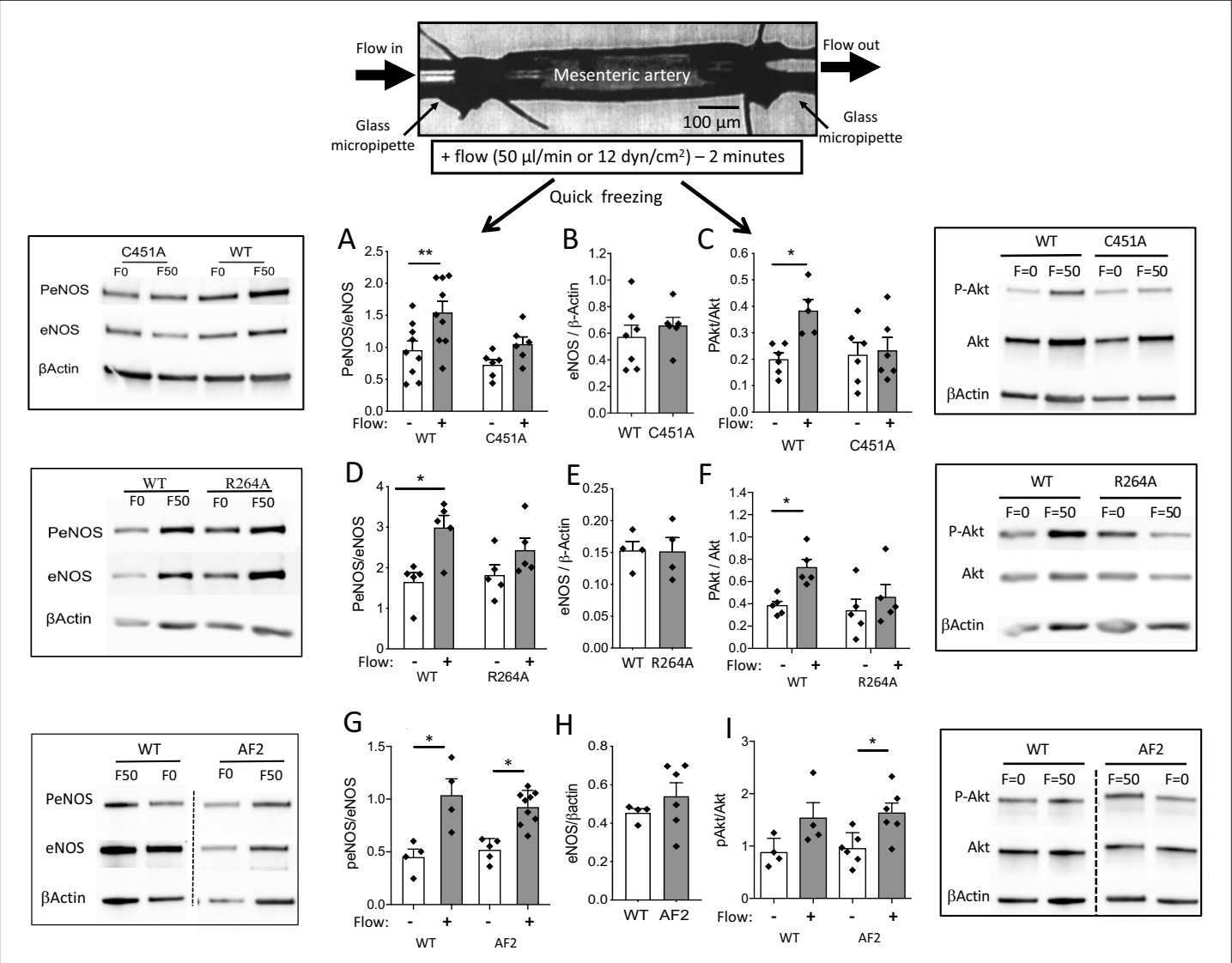

**Figure 5.** eNOS and Akt phosphorylation in response to flow in perfused isolated mesenteric resistance arteries. As illustrated on the scheme shown on the top of the figure, mesenteric resistance arteries were cannulated in vitro on glass micropipettes and perfused with physiological salt solution. Flow (50 µl/min or 12 dyn/cm²) was applied for 2 min before quick freezing of the artery. In control experiments no flow was applied. Western-blot analysis of eNOS, phospho (Ser1177)-eNOS (P-eNOS), Akt, phospho-Akt and β-actin in mesenteric arteries isolated from male C451A-ERα mice (C451A, **A to C**), R264A-ERα (R264A, **D to F**), AF2⁰ERα (AF2, **G to I**) and their littermate control (WT) was then performed. The ratio of P-eNOS / eNOS is shown in **A**, **D** and **G**. The expression level of eNOS/β-actin in unstimulated arteries is shown in **B**, **E** and **H**. The ratio of P-Akt / Akt is shown in **C**, **F** and **I**. Means ± the SEM are shown (n = 6 C451A-WT, n = 9 C451A-ERα, n = 5 R264A-ERα, n = 5 R264A-WT, n = 6 AF2⁰ERα and n = 4 AF2-WT mice). *p < 0.05 (panel **C**: p = 0.0374, panel **D**: p = 0.015, panel **F**: p = 0.0177, panel **G**: WT, p = 0.0234, AF2, p = 0.0465, panel **I**: p = 0.0152) and **p < 0.01 (panel **A**: p = 0.0045), two-tailed Mann-Whitney test. Data and analysis in *Figure 5—source data 1*.

The online version of this article includes the following figure supplement(s) for figure 5:

**Source data 1.** Data and statistical analysis from experiments plotted in *Figure 5A–I*.

**Source data 2.** All the blots for *Figure 5*.

The acute response to ATP and to the Piezo1 agonist YODA-1 were not affected by the absence of membrane ERα (C451A-ERα mice). In addition, the mechanosensitive channel blocker GsMTx4 similarly affected FMD in C451A-ERα and WT mice, suggesting that the defect in FMD is located downstream flow sensing (*Figure 7—figure supplement 3*).

This pharmacological approach suggests that reactive oxygen species could reduce FMD in arteries from C451A-ERα, while $H_2O_2$ could maintain in part the dilatory response induced by flow in the absence of membrane ERα.

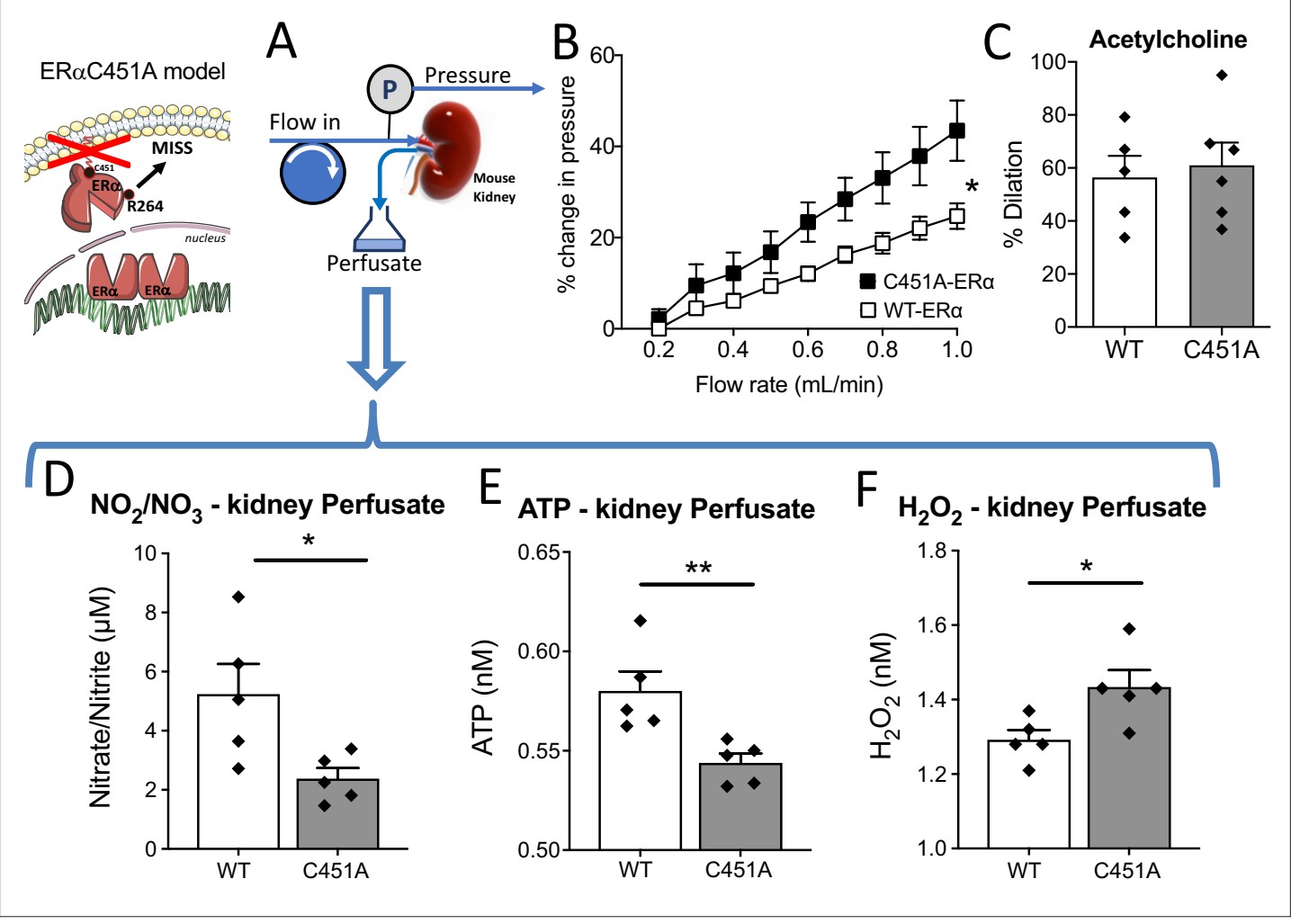

**Figure 6.** Isolated and perfused kidney from C451A-ERα mice. In the isolated and perfused kidney (**A**), the flow-pressure relationship was determined in C451A-ERα and WT mice (**B**). (**C**) Acetylcholine (1 μM)-mediated relaxation. The levels of nitrate-nitrite (**D**), ATP (**E**) and $H_2O_2$ (**F**) level were quantified in the perfusate collected from the kidney. Means ± the SEM are shown (n = 5 C451A-WT and 7 C451A -ERα mice). *p < 0.05, two-way ANOVA for repeated measurements (panel **B**, C451 vs WT: p = 0.0308 Interaction: p = 0.0008). Two-tailed Mann-Whitney tests (panels **C** to **F**: p > 0.999, p = 0.0317, p = 0.0079 and p = 0.0317, respectively). Data and analysis in *Figure 6—source data 1*.

The online version of this article includes the following figure supplement(s) for figure 6:

**Source data 1.** Data and statistical analysis from experiments plotted in *Figure 6B–F*.

## In vivo pharmacological ROS reduction restored FMD inC451A-ERα mice

As acute antioxidant drugs restored FMD in mesenteric arteries isolated from C451A-ERα mice, we further explored the involvement of oxidative stress in the alteration of FMD by the use of two different antioxidant treatments in vivo.

After 2 weeks of treatment with the antioxidant TEMPOL, there was no longer a discernible difference between the response of WT and C451A-ERα mice to flow (*Figure 8A*). Mice body weight, arterial diameter, phenylephrine-, and KCl-mediated contraction as well as acetylcholine-mediated dilation were not different between TEMPOL-treated WT and C451A-ERα mice (*Figure 8B–F*).

A similar pattern was observed in mice treated for 4 weeks with vitamin E and vitamin C with no difference in FMD between WT and C451A-ERα mice (*Figure 8G to L*).

Thus, antioxidant treatment normalized FMD in C451A-ERα mice to the level of FMD in WT mice. Altogether, these data suggest that the absence of membrane-associated ERα increases oxidative stress, which in turn could be responsible for a large part of the alteration of NO-dependent FMD.

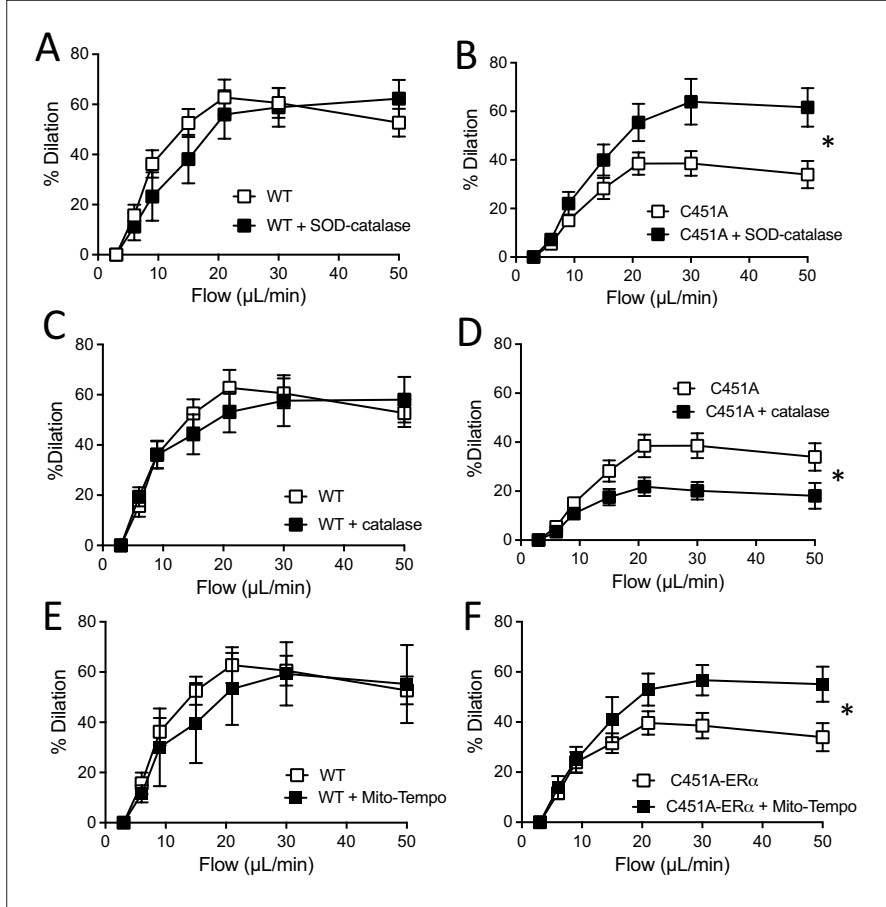

**Figure 7.** Flow-mediated dilation and oxidative stress. Flow-mediated dilation was determined in mesenteric resistance arteries isolated from male WT and C451A-ERα mice before and after addition of PEG-SOD and catalase (SOD-catalase, **A and B**), catalase (**C and D**) or Mito-Tempo (**E and F**). Flow rate was 3, 6, 9, 12, 15, 30, and 50 µl/min corresponding to 0.8, 1.2, 2, 2.8, 4, 8, and 12 dyn/cm$^2$. Means ± the SEM are shown (n = 3–9 mice per group, see details in *Figure 7—source data 1*). *p < 0.05, two-way ANOVA for repeated measurements (panel **A to F**: p = 0.5887, p = 0.0321, p = 0.7170, p = 0.0311, p = 0.7641 and p0.0354, respectively).

The online version of this article includes the following figure supplement(s) for figure 7:

**Source data 1.** Data and statistical analysis from experiments plotted in *Figure 7A–L*.

**Figure supplement 1.** Gene expression profile in the mesenteric isolated from mice lacking membrane-ERα.

**Figure supplement 1—source data 1.** Data from experiments plotted in *Figure 7—figure supplement 1A–X* and in *Figure 7—figure supplement 2A–T*.

**Figure supplement 2.** Gene expression profile in the mesenteric isolated from mice lacking membrane-ERα.

**Figure supplement 3.** Mechanosensitive channels and ATP in FMD.

**Figure supplement 3—source data 1.** Data and statistical analysis from experiments plotted in *Figure 7—figure supplement 3A–C*.

## Discussion

We report here that endothelial membrane-associated ERα contributed to optimize flow (shear stress)-mediated dilation in young healthy mouse resistance arteries in a ligand-independent manner.

Previous experimental studies have reported the vascular benefit of estrogens on blood flow homeostasis, and E2 improves endothelium-dependent relaxation when it is reduced in diseased conditions. (*Huang et al., 2001*; *Al-Khalili et al., 1998*; *Huang et al., 2000*; *Svedas et al., 2002*; *LeBlanc et al., 2009*). Nevertheless, no difference in FMD has been observed between healthy men and women (*Sullivan et al., 2015*). In agreement, our results show that acute (20 min) incubation with

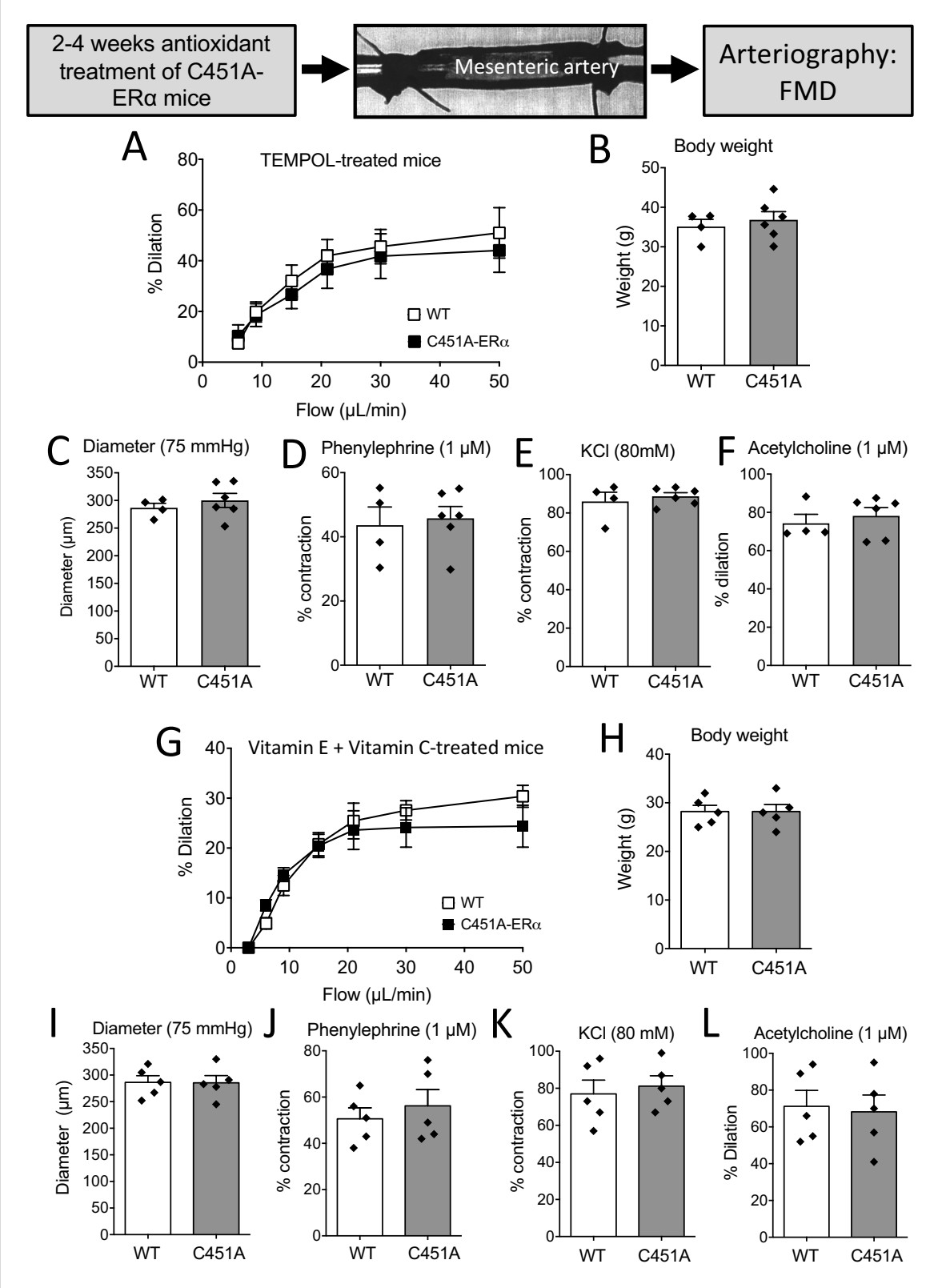

**Figure 8.** FMD after antioxidant treatments in mice lacking membrane-ERα. FMD was determined in mesenteric resistance arteries isolated from male WT and C451A-ERα mice treated for 2 weeks with the anti-oxidant TEMPOL (**A to F**) or with a combination of vitamin E and vitamin C for 4 weeks (**G to L**). At the end of the treatments arteries were collected and mounted in an arteriograph for the measurement of FMD (**A and G**), body weight (**B and H**), arterial diameter (**C and I**), phenylephrine (1 µM, **D and J**)- and KCl (80 mM, **E and K**)-mediated contraction and acetylcholine (1 µM)-mediated

*Figure 8 continued on next page*

*Figure 8 continued*

dilation (**F and L**). Flow rate was 3, 6, 9, 12, 15, 30, and 50 µl/min corresponding to 0.8, 1.2, 2, 2.8, 4, 8, and 12 dyn/cm². Means ± the SEM are shown (n = 4 C451A-WT and 6 C451A-ERα mice treated with TEMPOL and n = 5 mice per group treated with vitamin E and vitamin C). NS, two-way ANOVA for repeated measurements (panel **A**: p = 0.6345 and G: p = 0.6482). NS, Two-tailed Mann-Whitney tests (panels **B to F** and **H to L**). Data and analysis in *Figure 8—source data 1*.

The online version of this article includes the following figure supplement(s) for figure 8:

**Source data 1.** Data and statistical analysis from experiments plotted in *Figure 8A–L*.

exogenous E2, E4, or ICI-182780 did not affect FMD. Similarly, incubation with the GPER antagonist G-36 did not affect FMD, excluding ligand-activated GPER actions in FMD. In addition, endogenous estrogens in female WT mice had no impact on FMD as it was equivalent in male, female and ovariectomized female mice. These data suggest that FMD involves unliganded ERα activation in response to shear stress. In agreement, a recent study has reported another action of unliganded ERα, namely its inhibitory action on endothelial cell proliferation and migration (*Lu et al., 2017*).

Although FMD was reduced in *Esr1*[-/-] mice, agonist-mediated endothelium dependent (acetylcholine and insulin) and independent (SNP) dilation was not affected suggesting a selective reduction in flow (shear stress)-dependent signaling without a change in receptor-dependent dilation in the endothelium and in the smooth muscle.

The present study also showed that FMD involves membrane-associated ERα. FMD was similarly reduced in *Esr1*[-/-] mice and in both C451A-ERα and R264A-ERα mice. Although acute response (FMD and agonist-dependent dilation) can only be attributed to membrane-associated events, the expression level of the enzymes involved in the process could be modulated by the nuclear effects of ERα. Thus, endothelium-dependent dilation was investigated in mice lacking either the nuclear activating function AF2 of ERα or membrane-dependent action of ERα. Membrane-ERα is located at the level of the caveolae through either a binding to caveolin-1 or to striatin, thus creating a link with the Gαi and Gβγ proteins (*Arnal et al., 2017*). In order to abrogate the membrane effects of ERα, we used two different models. First, we used C451A-ERα mice that lack the palmitoylation site (cysteine at position 451) of the receptor so that the anchorage of ERα to the plasma membrane and the link to caveolin-1 is prevented (*Adlanmerini et al., 2014*). We also used a knock-in mouse model of ERα mutated for the arginine 264 (R264A-ERα mice) suppressing its interaction with Gαi involved in rapid eNOS activation (*Adlanmerini et al., 2020*). The fact that FMD was similarly reduced in *Esr1*[-/-], C451A-ERα, and R264A-ERα mice without change in receptor-dependent dilation, strongly supports that membrane-associated ERα is involved in FMD. This is further supported by the absence of reduction in FMD observed in AF2-ERα mice which only lack the AF2 nuclear function of ERα. AF2 is also involved in the vascular response to a chronic increase in flow (flow-mediated remodeling) which is absent in AF2⁰-ERα mice but fully present in C451A-ERα and R264A-ERα mice (*Guivarc'h et al., 2018*). This remodeling is a chronic adaptation of the vascular wall associated with changes in arterial diameter and wall mass within 2 weeks after a chronic rise in blood flow in vivo (*Chehaitly et al., 2021*). A chronic increase in blood supply, such as that needed for collateral growth in ischemic disorders, induces an increase in diameter together with wall thickening so that both shear and tensile stress are normalized within 1 week following the flow increase (*Silvestre et al., 2001*). This remodeling involves an early inflammatory phase allowing cell growth and reorganization in the arterial wall (*Caillon et al., 2016*) and a dilatory phase involving NO, prostaglandins and CO production (*Dumont et al., 2007*; *Belin de Chantemèle et al., 2010*; *Freidja et al., 2011*). Noteworthy, flow-mediated remodeling is absent in ovariectomized rats and mice and in *Esr1*[-/-] mice (*Tarhouni et al., 2013*) whereas this remodeling is preserved in ovariectomized rats treated with E2 (*Tarhouni et al., 2014b*) or resveratrol (*Petit et al., 2016*). More recently, we have shown that this remodeling requires activation of AF2 and is independent on membrane-located ERα (*Guivarc'h et al., 2018*).

Another membrane receptor for E2 located at the plasma membrane is GPER (*Prossnitz and Barton, 2011*). Both ligand-dependent and ligand-independent activation of GPER have been reported (*Meyer et al., 2016*). GPER is involved in regulation of reproductive functions, endocrine regulation and metabolism, cardiovascular, kidney, neuroendocrine and cerebral functions function as well as immune cell function. Furthermore, previous studies suggest a role for GPER in hypertension, kidney diseases, diabetes, and immune diseases. Consequently, GPER is a potential therapeutic target for the treatment of these diseases (*Prossnitz and Barton, 2011*). In the present study,

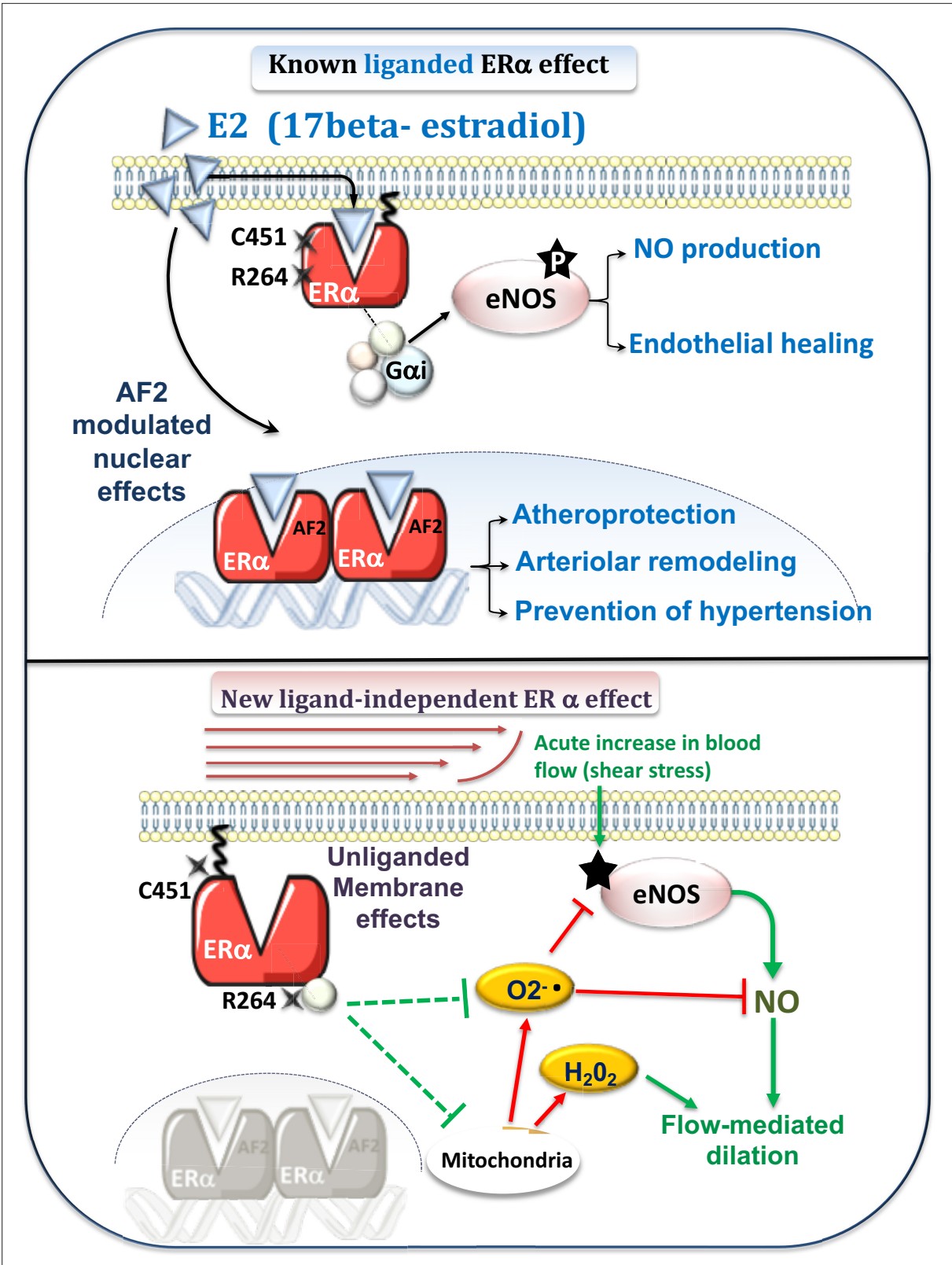

**Figure 9.** Schematic representation of the known E2-mediated ERα-dependent protective effects (upper panel) and of the new pathways described in the present study (lower panel). Previous works (upper panel) have demonstrated the role of E2 and the nuclear activating function AF2 of ERα against atherosclerosis and hypertension (*Guivarc'h et al., 2018*) as well as in flow-mediated outward remodeling (*Tarhouni et al., 2013*; *Guivarc'h et al., 2018*). E2-stimulated membrane-located ERα is involved in E2-dependent NO production and in endothelial healing (*Adlanmerini et al., 2014*). New

*Figure 9 continued on next page*

*Figure 9 continued*

pathway described in the present work (lower panel): Flow, by stimulation of the surface of the endothelial cell by shear stress, activates the NO pathway (e.g. phosphorylation of eNOS: P-eNOS). This results in the production of NO, which in turn induces relaxation of the smooth muscle and thus dilation. In parallel, flow activates membrane-associated ERα, which reduces oxidative stress ($O_2^-$. and $H_2O_2$) due to NADPH-oxidase activity or of mitochondrial origin. This results in enhanced NO bioavailability. The absence of membrane-associated ERα could lead to the production of $O_2^-$., which attenuates NO-dependent dilation despite a remaining dilation due to a rise in $H_2O_2$ production.

incubation with the GPER antagonist G-36 did not affect FMD, ruling out the role of ligand-activated GPER in FMD. However, a possible role of unliganded GPER activation cannot be excluded in case of a crosstalk between membrane-dependent ERα and GPER activation.

To characterize the effect of membrane-ERα on the NO pathway which is involved in FMD, we investigated the effect of L-NNA-mediated inhibition of NO-synthesis on FMD. L-NNA inhibited FMD in arteries from WT and AF2$^0$ERα, but not arteries from *Esr1*$^{-/-}$, C451A-ERα and R264A-ERα mice, suggesting that membrane-associated ERα is involved in NO-dependent FMD. This is in agreement with a previous study that used the ERα Neo-KO model with incomplete deletion, thereby showing that the NO pathway (dilation sensitive to L-NAME) was reduced in response to flow in the gracilis artery of male mice (*Sun et al., 2007*). In contrast to FMD, L-NNA strongly reduced acetylcholine-mediated dilation in WT, *Esr1*$^{-/-}$ AF2$^0$ERα, C451A-ERα and R264A-ERα mice. Thus, the NO-pathway can be activated in response to receptor stimulation in C451A-ERα and R264A-ERα mice, whereas only its activation by flow was reduced in these mice lacking only membrane-ERα signaling. FMD can involve prostaglandins and EDHF such as EETs (*Sun et al., 2007*) and EDHF was shown to mediate estrogen-mediated dilation of the uterine arteries (*Burger et al., 2009*). Nevertheless, in the present study, cyclooxygenase inhibition with indomethacin and EETs production inhibition with MSPPOH did not affect FMD in both WT and *Esr1*$^{-/-}$, AF2$^0$ERα, C451A-ERα and R264A-ERα mice, suggesting a limited role of the pathway in mesenteric arteries of male mice. Consequently, the remaining FMD following the addition of L-NNA, indomethacin and MSPPOH relies probably on other hyperpolarizing agents. Indeed, endothelium-dependent hyperpolarization (EDH) has a major role in resistance arteries homeostasis (*Brandes et al., 2000*; *Garland and Dora, 2017*) and COX-derivatives can also induce EDH in resistance arteries and in the carotid when submitted to flow (*Ohlmann et al., 2005*; *Bergaya et al., 2001*). In humans, FMD measured in the brachial artery relies mainly on the production of NO (*Alexander et al., 2021*). Furthermore, changes in FMD in the brachial artery predict well the endothelial dysfunction in human resistance arteries (*Park et al., 2001*). Nevertheless, the difference in the nature of the agents involved in FMD besides NO between humans and mice could be a limitation of the present study. As stated above, the involvement of EDH in FMD is greater in mouse resistance arteries than in humans when measured at the level of the brachial artery.

As FMD has a key role in blood flow delivery to organs (*Hill et al., 2010*), we investigated the flow-pressure relationship in the mouse kidney. In agreement with the reduction in FMD observed in resistance arteries, we found a leftward shift of the flow-pressure relationship in C451A-ERα mice further confirming the decreased sensitivity to flow of the resistance vasculature. Recent studies have shown that flow activates Piezo1-dependent release of ATP through pannexin hemi-channel followed by P2Y2 activation and NO production by endothelial cells (*Wang et al., 2015*; *Wang et al., 2016*). We found that both nitrate-nitrite and ATP productions were lower in the kidney perfusate from C451A-ERα than in WT mice, in agreement with the reduced NO-dependent FMD (sensitive to L-NNA) observed in isolated arteries. As in mesenteric arteries, we have previously shown that eNOS expression level in the kidney is not altered by the absence of membrane ERα in C451A-ERα (*Guivarc'h et al., 2020*). Although NO and ATP production were reduced in kidneys from C451A-ERα, the acute response to ATP and to the Piezo1 agonist YODA-1 was not affected by the absence of membrane ERα. In addition, the mechanosensitive channel blocker GsMTx4 similarly affected FMD in C451A-ERα and WT mice, suggesting that the defect in FMD associated with the absence of membrane ERα is probably located downstream flow sensing.

The reduction in NO-dependent FMD found in arteries from mice lacking membrane-ERα could be due to an excessive ROS production as the chronic treatment of C451A-ERα mice with an antioxidant treatment restored FMD to control level. In agreement, we found that $H_2O_2$ production by the kidney was higher in C451A-ERα than in WT mice. This observation is in agreement with a previous work that has shown that E2 increases the release of bioactive NO by inhibition of superoxide anion production

in bovine endothelial cells (**Arnal et al., 1996**). Accordingly, we found that reducing total ROS or mitochondrial ROS production improved FMD in C451A-ERα mice. By contrast, catalase which eliminates $H_2O_2$ reduced FMD in C451A-ERα mice suggesting that an excessive ROS production due to the absence of membrane-associated ERα had a dual effect with (1) a reduction of NO bioavailability and (2) an increase in $H_2O_2$ production contributing to some vasodilating effect. Noteworthy, kidney perfusates showed higher levels of $H_2O_2$ in C451A-ERα mice. An excessive ROS production could also alter eNOS activation as previously shown through increased phosphatase activation (**Ding et al., 2020**). Previous studies have also shown that shear stress induces a more quiescent and less oxidative phenotype in endothelial cells (**Doddaballapur et al., 2015**; **Wu et al., 2018**), thus reducing the oxidative products of mitochondrial origin. Nevertheless, in a context of known dysfunctional FMD, a previous work has shown that $H_2O_2$ could mediate FMD in human coronary arteries from patients suffering coronary artery disease although $H_2O_2$ remains deleterious (**Freed et al., 2014**).

## Conclusion

To conclude, these data demonstrate for the first time a major role of ERα, and more precisely of non liganded endothelial membrane-located ERα for optimal FMD and thereby a potential role in local blood flow homeostasis. The mechanism appears to involve an optimization of NO activation and/or a decrease in ROS production as depicted in **Figure 9**. The functional consequences in terms of arteriolar and tissue protection should now be investigated.

# Materials and methods

**Key resources table**

| Reagent type (species) or resource | Designation | Source or reference | Identifiers | Additional information |
|---|---|---|---|---|
| Strain, strain background (*Mus musculus,* males and females) | *Esr1-/-* C57BL/6 J (Symbol: Esr1tm1.1Mma, Synonyme: ERalpha Knockout) | Mouse Clinical Inst., Strasbourg, France, **Dupont et al., 2000** | | MGI:2386760 |
| Strain, strain background (*Mus musculus,* males) | AF2°ER α , C57BL/6 J (Symbol: Esr1tm1.1Ohl Synonym: ERalpha-AF2°) | Mouse Clinical Inst., Strasbourg, France, **Billon-Galés et al., 2009** | | MGI:4950046 |
| Strain, strain background (*Mus musculus,* males) | C451A-ER α , C57BL/6 N (Symbol: Esr1tm1.1Ics Synonyme: C451A-ERalpha knock-in) | Mouse Clinical Inst., Strasbourg, France, **Adlanmerini et al., 2014** | | MGI:5574591 |
| Strain, strain background (*Mus musculus,* males) | *Tek*Cre/+:*Esr1*f/f, C57BL/6 (B6.Cg-Tg(Tek-cre)12Flv/J backcrossed with *Esr1*tm1.2Mma Synonym: *Tie2*Cre *ERα*lox/lox) | *Esr1*lox/lox: Mouse Clinical Institut, Strasbourg, France. *Tek*Cre: Jackson Lab (Bar Harbor, Me), **Billon-Galés et al., 2009** *Tek*Cre: **Koni et al., 2001** *Esr1*lox/lox: **Dupont et al., 2000** | *Tek*Cre: *Esr1*lox/lox: | MGI:3775510 |
| Strain, strain background (*Mus musculus,* males) | *Esr2-/-,* C57BL/6 J (Symbol: Esr2tm1Mma Synonym: ERbeta) | Mouse Clinical Inst., Strasbourg, France, **Dupont et al., 2000** | | MGI:2386761 |
| Strain, strain background (*Mus musculus,* males) | R264A-ER α , C57BL/6 N | Mouse Clinical Inst., Strasbourg, France, **Adlanmerini et al., 2020** | | No MGI ID yet |
| Antibody | Anti-eNOS, (mouse monoclonal, clone3) | BD Biosciences | Cat# 610297, RRID:AB_397691 | WB (1:1000) |
| Antibody | Anti-phospho-eNOS, pS1177 (Mouse monoclonal,Clone 19/eNOS/S1177) | BD Biosciences | Cat# 612392, RRID:AB_399750 | WB (1:1000) |
| Antibody | Anti-beta-actin, (Mouse monoclonal, clone AC-74) | Sigma-Aldrich | Cat#: 5316; RRID:AB_476743 | WB (1:5000) |

*Continued on next page*

*Continued*

| Reagent type (species) or resource | Designation | Source or reference | Identifiers | Additional information |
|---|---|---|---|---|
| Antibody | Anti-Akt Pan, (rabbit monoclonal, clone C67E7) | Cell signalling technology Ozyme | Cat#: 4691; RRID:AB_915783 | WB (1:1000) |
| Antibody | Anti-phospho-Akt, S473, (rabbit monoclonal, clone D9E) | Cell signalling technology Ozyme | Cat#: 4060; RRID:AB_2315049 | WB (1:2000) |
| Antibody | Anti-mouse IgG (H + L) Secondary antibody HRP (Goat polyclonal) | Thermo scientific | Cat#: 31430; RRID:AB_228307 | WB (1:5000) |
| Antibody | Anti-rabbit IgG(H + L) Secondary antibody HRP (Goat polyclonal) | Thermo scientific | Cat#: 31460; RRID:AB_228341 | WB (1:10000) |
| Chemical compound, drug | vitamin C | Sigma Aldrich Merck, *Favre et al., 2011* | A5960 | |
| Chemical compound, drug | vitamin E | Sigma Aldrich Merck, *Favre et al., 2011* | T3251 | |
| Chemical compound, drug | Mito-tempo | Sigma Aldrich Merck, *Freed et al., 2014* | SML0737 | |
| Chemical compound, drug | catalase | Sigma Aldrich Merck, *Bouvet et al., 2007* | C3155 | |
| Chemical compound, drug | PEG-superoxide dismutase (SOD) | Sigma Aldrich Merck, *Bouvet et al., 2007* | S9549 | |
| Chemical compound, drug | Estetrol (E4) | Sigma Aldrich Merck, *Abot et al., 2014* | SML1523 | |
| Chemical compound, drug | ICI 182 780 | Tocris Biotechne, *Meyer et al., 2010* | 1047 | |
| Chemical compound, drug | G-1 ((±)–1-[(3a$R$*,4$S$*,9b$S$*)–4-(6-Bromo-1,3-benzodioxol-5-yl)–3 a,4,5,9b-tetrahydro-3$H$-cyclopenta[$c$]quinolin-8-yl)]- ethanone | Cayman chemical Bertin Bioreagent, *Meyer et al., 2010* | 10008933 | |
| chemical compound, drug | G-36 ((±)-(3a$R$*,4$S$*,9b$S$*)–4-(6-Bromo-1,3-benzodioxol-5-yl)–3 a,4,5,9b-tetrahydro-8-(1-methylethyl))–3$H$-cyclopenta[$c$]quinoline | Cayman chemical Bertin Bioreagent, *Meyer et al., 2016* | 14,397 | |
| Sequence-based reagent | N-(methylsulfonyl)–2-(2-propynyloxy)-benzenehexanamide (MSPPOH) | Cayman chemical Bertin Bioreagent, *Dietrich et al., 2009* | 75,770 | |
| Chemical compound, drug | Grammostola spatulata mechanotoxin 4 (GsMTx4) | Alomone Labs, *John et al., 2018* | STG-100 | |
| Chemical compound, drug | YODA1 | Bertin Bioreagent, *Lhomme et al., 2019* | SML1558 | |
| Chemical compound, drug | ATPγS | Tocris Biotechne, *Kukulski et al., 2009* | 4080 | |
| Chemical compound, drug | 4-hydroxy-2,2,6,6-tetramethylpiperidine (TEMPOL) | Sigma Aldrich Merck, *Freidja et al., 2014* | 176,141 | |
| commercial assay or kit | Nitric oxide metabolite detection kit | Cayman Chemical | 780,051 | |
| commercial assay or kit | Hydrogen peroxide assay kit | Abcam | Ab102500 | |
| commercial assay or kit | ATP determination kit | Invitrogen Molecular Probes | A22066 | |

## Animal protocol

We used 5–6 month-old male mice lacking the gene encoding ERα (*Esr1⁻/⁻*) (*Antal et al., 2008*) or ERβ (*Esr2⁻/⁻*) (*Antal et al., 2008*), mice lacking ERα selectively targeted to the endothelium (Tek[Cre/+]:ER[f/f]) (*Toutain et al., 2009*), mice lacking the nuclear activation function AF2 (AF2[0]ERα mice) (*Billon-Galés et al., 2011*), mice in which the codon for the cysteine (Cys451) palmitoylation site of ERα had been mutated to alanine (C451A-ERα mice) (*Adlanmerini et al., 2014*) and mice mutated for the arginine

264 of ERα (R264A-ERα mice) (*Adlanmerini et al., 2020*). Littermate +/+ mice were used as controls (designated wild-type, WT, or +/+) in each group.

In a separate series of experiments, 5–6 month-old female *Esr1*[-/-] and *Esr1*[+/+] mice were used for FMD measurements. The mice had been ovariectomized or left intact (with only a sham surgery), as previously described (*Toutain et al., 2009*).

In another series of experiments, 5–6 month-old male C451A-ERα and C451A-WT mice were treated with the antioxidant 4-hydroxy-2,2,6,6-tetramethylpiperidine (TEMPOL, 10 mg/kg per day, 2 weeks in drinking water) (*Belin de Chantemèle et al., 2009*) or with the antioxidants vitamin E (1 % in chow) and vitamin C (0.05 % in water) for 4 weeks. (*Favre et al., 2011*; *Contreras-Duarte et al., 2018*).

The mice were anesthetized with isoflurane (2.5%) and euthanized with $CO_2$. The mesentery and the uterus were quickly removed and placed in ice-cold physiological salt solution (PSS) (*Tarhouni et al., 2013*). Several segments of second-order arteries were collected for the functional study and for biochemical studies.

The experiments complied with the European Community standards for the care and use of laboratory animals and the Guide for the Care and Use of Laboratory Animals published by the US National Institutes of Health (NIH Publication No. 85–23, revised in 1996). The protocol was approved by the regional ethics committee (permits #14335, #16740, and #16108).

## Flow-mediated dilation in mesenteric arteries in vitro

Arterial segments, with internal diameters of approximately 200 µm, were cannulated at both ends on glass micro-cannulas and mounted in a video-monitored perfusion system (Living System, LSI, Burlington, VT, USA) (*Iglarz et al., 1998*; *Bolla et al., 2002*). Individual artery segments were bathed in a 5 ml organ bath containing PSS (pH: 7.4, $pO_2$: 160 mmHg, and $pCO_2$: 37 mmHg) and perfusion of the artery was carried out with two peristaltic pumps, one controlling the flow rate and the other under the control of a pressure-servo control system. The pressure was set at 75 mmHg and flow (3–50 µl per min) was generated through the distal pipette with a peristaltic pump. Flow steps were 3, 6, 9, 12, 15, 30 and 50 µl/min which correspond to 0.8, 1.2, 2, 2.8, 4, 8 and 12 dyn/cm$^2$.

FMD was determined before and after pretreatment with N(omega)-nitro-L-arginine (L-NNA, 100 µM, 30 min), L-NNA plus indomethacin and then with L-NNA plus indomethacin (10 µmol/L) plus N-(methylsulfonyl)–2-(2-propynyloxy)-benzenehexanamide (MSPPOH, 10 µmol/L).

In a separate series of experiments, the effect the mechanosensitve ionic channels blocker Grammostola spatulata mechanotoxin 4 (GsMTx4) (5 µmol/L, delivered intraluminally and incubated for 45 min) (*John et al., 2018*).

The impact of ex vivo modulation of ERα on FMD was evaluated after 20 minutes of incubation with the ERα agonists E2 (10 nM) or E4 (1 µmol/L), the GPER agonist G-1 (1 µmol/L) (*Meyer et al., 2010*), the GPER antagonist G-36 (1 µmol/L) (*Yu et al., 2018*) or the estrogen receptor downregulator and GPER agonist ICI 182 780 (0.1 µmol/L) (*Meyer et al., 2010*).

In another series of experiments, FMD was measured before and after incubation (20 min) of the arteries with PEG-superoxide dismutase (SOD, 120 U/mL) plus catalase (80 U/mL) (*Bouvet et al., 2007*), catalase (80 U/mL), or Mito-Tempo (1 µmol/L) (*Freed et al., 2014*).

## Pharmacological profile of isolated mesenteric arteries

Segments of mesenteric arteries were mounted in a wire-myograph (Danish Myo Technology, Denmark) as previously described (*Loufrani et al., 2002*) in order to obtain cumulative concentration-response curves (CRCs) to acetylcholine (ACh) before and after pretreatment with L-NNA (10 µmol/L) and then with L-NNA (100 µmol/L) plus indomethacin (10 µmol/L).

In a separate series of experiments, CRCs to YODA1, ATPγS (ATP) were performed.

Prior to each CRC, the arteries were submitted to phenylephrine to obtain approximately 50 % of the maximal contractile response of the vessel assessed by KCl (80 mM)-mediated contraction at the beginning of the experiment.

## Western-blot analysis

Arterial segments were cannulated under pressure (75 mmHg), and flow (50 µl/min) was applied after precontraction with phenylephrine (1 µM). After 2 minutes, the arteries were quick-frozen. Due to the limited size of the resistance arteries segments were pooled before analysis. Protein expression (eNOS, phospho-eNOS, Akt and phospho-Akt) was then determined using Western blot (*Bouvet et al., 2007*).

## Perfused isolated mouse kidney

In a separated series of experiments, the right renal artery was cannulated in anesthetized mice and the kidney was excised and perfused at 37 °C with PSS as previously described (*Begorre et al., 2017*). The right renal artery was cannulated in anesthetized mice (as described above) with a polyethylene catheter (PE-10, 0.28 mm internal diameter, 0.61 mm external diameter, Intramedic, Evry, France). The kidney was then excised and perfused without interruption of kidney flow at 37 °C with PSS. The perfusion solution was dialyzed and the pH was adjusted to 7.4. Perfusion rate was 600 µl/min and perfusion pressure was measured continuously (PT-F pressure transductor, Living System, Burlington, VT). Endothelium-mediated dilation was tested using ACh (1 µmol/L) after precontraction with Phe (1 µmol/L). Flow-pressure relationship was assessed through an stepwise increase in perfusion flow associated with the continuous measurement of the perfusion pressure.

The PSS perfusing the kidney (perfusate in the scheme shown in *Figure 6*) was collected in baseline conditions (flow = 600 µl/min) and immediately frozen in liquid $N_2$ and then stored at –80 °C.

## Determination of nitrate and nitrite, ATP and $H_2O_2$ levels in the kidney perfusate

To determine flow-induced nitrate-nitrite, ATP or $H_2O_2$ release from the perfused mouse kidney, 500 µl of perfusate was collected. The perfusate was then centrifuged 10 min at 14 000 rpm and the supernatant added into a spin column with 10 kDa molecular weight cut-off filter for ultrafiltration (10KD Spin Column Abcam ab93349) and centrifuged at 10 000 rpm for 10 min. The centrifuged solutions was then used for nitrate-nitrite, ATP and $H_2O_2$ measurement.

Nitrate and nitrite levels in kidney perfusate were determined using a nitrate/nitrite fluorometric assay kit from Cayman Chemical (Nitric Oxide Metabolite Detection Kit Nb°780051) according to the manufacturer's instructions.

Hydrogen peroxyde ($H_2O_2$) level was determined using the fluorimetric method of a hydrogen peroxide assay kit from Abcam (Hydrogen Peroxide Assay Kit Colorimetric/Fluorometric NbAb102500) according to the manufacturer's instructions.

ATP level was measured using ATP determination kit from Invitrogen Molecular Probes (ATP Determination Kit Nb A22066) according to the manufacturer's instructions.

## Preparation of endothelial cells enriched fraction for transcriptional analysis

Endothelial cells enriched fractions were obtained as previously described (*Briot et al., 2014*). Briefly, 5-week-old female mice were perfused with PBS. The descending thoracic aorta was dissected and perfused with RLT buffer (Qiagen, Valencia, CA) containing 1 % beta-mercaptoethanol. Endothelial cells enrichment was confirmed by the increased endothelial marker *Tek* expression level and the absence of smooth muscle cell marker *Cnn1* (*Kalluri et al., 2019*) compared to the total aorta.

## Evaluation of gene expression by quantitative real-time PCR in mesenteric arteries

Gene expression was investigated using quantitative polymerase chain reaction after reverse transcription of total RNA (RT-qPCR). Mesenteric arteries were stored at −20 °C in RNAlater Stabilization Reagent (Qiagen, Valencia, CA, USA) until use. RNA was extracted using the RNeasy Micro Kit (Qiagen, Valencia, CA, USA) following manufacturer instructions. RNA extracted (300 ng) was used to synthesize cDNA using the QuantiTect Reverse Transcription Kit (Qiagen, Valencia, CA, USA). RT-qPCR was performed with Sybr Select Master Mix (Applied Biosystems Inc, Lincoln, CA, USA) reagent using a LightCycler 480 Real-Time PCR System (Roche, Branchburg, NJ, USA). Primer sequences are shown in the *Supplementary file 1*. *Gapdh*, *Hprt* and *Gusb* were used as housekeeping genes. Analysis was not performed when Ct values exceeded 35. Results were expressed as: $2^{(Ct\ target-Ct\ housekeeping\ gene)}$.

## Statistical analysis

The results are expressed as means ± the SEM. The significance of the differences between groups was determined by analysis of variance (two-way ANOVA for consecutive measurements) followed by Bonferroni's test for the FMD and the agonist-mediated concentration-response curves. A two-tailed Mann-Whitney test (when comparing two groups) or a Kruskal-Wallis test (more than two groups) was used for the other comparisons as indicated in the figure legends. Probability values less than 0.05 were considered significant.

## Acknowledgements

*Esr2*$^{-/-}$, *Esr1*$^{-/-}$ and AF2$^0$ERα mice and the corresponding littermate WT mice were kindly provided by Prof. P Chambon and Dr. A Krust (Institute of Genetics and Molecular and Cellular Biology, Strasbourg, France). We thank J.M. Foidart and Mithra Pharma for providing Estetrol (E4). This work was supported in part by the foundation for Medical Research (*Fondation pour la Recherche Médicale*, Mitoshear project, contract FRM - DPC20171138957), the National Agency for Research (*Agence Nationale de la Recherche*, Estroshear project, contract # ANR-18-CE14-0016-01), INSERM, University of Toulouse III, the Fondation pour la Recherche Médicale (Equipe FRM DEQ20160334924), the Fondation de France (contract 00086486), the Région Occitanie, and the Institut Universitaire de France in Toulouse. JF was supported by the Lefoulon-Delalande Foundation. ChF and JR were supported by the National Agency for Research (*Agence Nationale de la Recherche*, Estroshear project, contract # ANR-18-CE14-0016-01).

## Additional information

### Funding

| Funder | Grant reference number | Author |
| --- | --- | --- |
| Fondation pour la Recherche Médicale | FRM - DPC20171138957 | Daniel Henrion |
| Fondation pour la Recherche Médicale | Equipe FRM DEQ20160334924 | Jean-François Arnal |
| Agence Nationale de la Recherche | ANR-18-CE14-0016-01 | Chanaelle Fébrissy Jordan Rivron |
| Fondation Lefoulon Delalande | | Julie Favre |
| Institut National de la Santé et de la Recherche Médicale | | Daniel Henrion Jean-François Arnal |
| Université de Toulouse | | Jean-François Arnal |
| Fondation de France | 00086486 | Jean-François Arnal |
| Région Occitanie Pyrénées-Méditerranée | | Jean-François Arnal |
| Institut Universitaire de France | | Jean-François Arnal |

The funders had no role in study design, data collection and interpretation, or the decision to submit the work for publication.

### Author contributions

Julie Favre, Conceptualization, Data curation, Investigation, Methodology, Resources, Validation, Writing – original draft, Writing – review and editing; Emilie Vessieres, Data curation, Formal analysis, Methodology; Anne-Laure Guihot, Coralyne Proux, Investigation, Methodology; Linda Grimaud, Data curation, Methodology, Resources; Jordan Rivron, Investigation, Methodology, Validation;

Manuela CL Garcia, Investigation, Validation; Léa Réthoré, Investigation; Rana Zahreddine, Investigation, Validation, Writing – original draft; Morgane Davezac, Data curation, Methodology, Performed experiments requested for the revised manuscript; Chanaelle Fébrissy, Investigation, Validation, Visualization; Marine Adlanmerini, Conceptualization, Investigation, Validation; Laurent Loufrani, Formal analysis, Investigation, Supervision, Validation, Writing – original draft; Vincent Procaccio, Methodology, Project administration, Validation; Jean-Michel Foidart, Methodology, Resources; Gilles Flouriot, Conceptualization, Methodology, Validation; Françoise Lenfant, Conceptualization, Methodology, Supervision, Validation, Writing – original draft, Writing – review and editing; Coralie Fontaine, Methodology, Validation, Writing – original draft; Jean-François Arnal, Conceptualization, Funding acquisition, Writing – original draft; Daniel Henrion, Conceptualization, Funding acquisition, Methodology, Project administration, Validation, Writing – original draft

**Author ORCIDs**
Laurent Loufrani (iD) http://orcid.org/0000-0003-3397-2335
Daniel Henrion (iD) http://orcid.org/0000-0003-1094-0285

**Ethics**
The investigation was conducted in accordance with the guidelines from Directive 2010/63/EU of the European Parliament for the protection of animals used for scientific purposes (authorization of the laboratory: # 00577). The protocol was approved by the Institutional Animal Care and Use Committee (IACUC): Committee on the Ethics of Animal Experiments (CEAA) of "Pays de la Loire" (permits #14335, #16740, and #16108). The mice were anesthetized with isoflurane (2.5%) and euthanized using a $CO_2$ chamber and every effort was made to minimize suffering.

**Decision letter and Author response**
Decision letter https://doi.org/10.7554/eLife.68695.sa1
Author response https://doi.org/10.7554/eLife.68695.sa2

# Additional files

**Supplementary files**
• Supplementary file 1. Primer sequences used for the RT-qPCR.

• Transparent reporting form

**Data availability**
All data generated or analysed during this study are included in the manuscript and supporting files. Source data files are provided for each figure and supplement.

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
