## [Editor Report]

Using multiple genetically modified mouse models, the authors have demonstrated a novel role of membrane associated estrogen receptor alpha (ERα) signaling to modulate flow-mediated dilation (FMD) in a ligand-independent manner. Specifically, the results indicate that non-nuclear actions of membrane estrogen receptor α in endothelial cells support flow-mediated vasodilatation in animals of both sexes via mechanisms that are independent of estrogenic ligands, involving NO production and an attenuation of the NO-inactivating effects of reactive oxygen species. These findings highlight a novel role of ligand-independent activation of membrane estrogen receptor α in regulation of vascular physiology and possibly in disease, adding to the recently introduced paradigm shift in the understanding of estrogen and estrogen receptor function.

---

## [Decision Letter]

**Decision letter after peer review:**

Thank you for submitting your article "Membrane estrogen receptor alpha (ERα) optimizes flow-mediated dilation in both sexes, in a ligand-independent manner" for consideration by *eLife*. Your article has been reviewed by 4 peer reviewers, one of whom is a member of our Board of Reviewing Editors, and the evaluation has been overseen by a Senior Editor. The following individual involved in review of your submission has agreed to reveal their identity: Philip W Shaul (Reviewer #4).

The reviewers have discussed their reviews with one another, and this letter is to help you prepare a revised submission.

Essential revisions:

1) The ligand-independent activation of membrane ERα is a novel finding. However, simply ignoring a potential role of GPER in such ligand-independent regulation of FMD is a major flaw. The authors should carry out some straight-forward studies to look at the effects of pharmacological inhibition using GPER antagonist, G-36, which turns out to not only inhibit GPER but also via inhibition of constitutive GPER expression, to reduce abundance (and activity) of Nox1. The authors are advised to perform in vitro studies of vessels from WT mice, evaluating dilation in response to flow +/- GPER antagonist, as long as a parallel control study is done evaluating dilation in response to GPER agonist +/- GPER antagonist.

2) In the Introduction section, the authors are encouraged to cite original works related to non-nuclear signaling by the subpopulation of ER associated with the plasma membrane in endothelium.

3) In the Introduction section, the background information about cardiovascular disease risk in men versus women and the impact of estrogens on risk is not as clear-cut as the authors imply. A more balanced presentation of the clinical evidence is warranted.

4) In the Introduction, a more detailed account of the published literature in the field including studies in humans should be provided The Authors may consider citing some studies from among PMID 19126786; 7642876; 20185791; 26471832; 9176294; 17367797; 28645860; 26734763; 9791075; 16131583; 18319309; 19086257; 9930647.

5) In the Results section, Page 6, line 106 incorrectly refers the reader to Figure 1B.

6) In the Results section. Page 6, line 117: fulvestrant is an ERα and ERβ antagonist (pKi = 9 for both) as well as a GPER full agonist (pKi = 7).

7) In the Results section, Figures 2B, 2F and 2J: ERα protein abundance in endothelial cells should be evaluated, even if only feasible from vasculature that would yield more endothelial cells than mesenteric resistance arteries.

8) In the Results section, Figure 5B: the flow-pressure relationship was shifted left, or up, in C451A-ERα mice versus WT according to the group labeling, and not to the right.

9) In the Results section Page 17, line 281: the use of the term autoregulation here may be confusing, and more specific interpretation would be helpful (and better placed in the Discussion section).

10) In the Results section, how does flow increase eNOS Ser1177 phosphorylation and alter H_2_O_2_ via ERα mechanistically ? Are flow-related changes in Akt phosphorylation in endothelium altered in the absence of ERα?

11) In the Results section, Do non-nuclear actions of plasma membrane-associated ERα influence endothelial cell production of prostaglandins in the setting of FMD?

12) In the Results section, do non-nuclear actions of plasma membrane-associated ERα influence mechanosensitive ion channel localization or function in endothelial cells?

13) In the Results section, why was the antioxidant TEMPOL specifically selected? Would the Authors expect a similar outcome following treatment with other antioxidants such as e.g. quercetin or dimethyl fumarate?

14) In the Results section, does Tempol or PEG-catalase acutely restore FMD ex vivo?

15) In the Results section, Line 312: Chronic for a 2-week treatment does not sound very appropriate.

16) In the Results section, flow rate may be changed to shear stress in Dyn/cm2: this would help comparing between published works in the topic.

17) The Discussion section should be shortened, providing more succinct focused discussion of the interpretation of the findings, their implications, and possible explanations to fill the new knowledge gaps that result from the work.

18) In the Conclusion section, Figure 7: the schematic is only in part helpful because how non-nuclear actions of membrane-associated ERα in endothelial cells govern eNOS and ROS in response to flow is not addressed.

*Reviewer #1:*

Flow mediated dilation (FMD) is a response to acute shear stress and its reduction is known to be a hallmark of endothelium dysfunction, associated with aging and with cardiovascular and metabolic disorder. Estrogen receptor alpha (ERα) was previously found to be associated with FMD in mouse and human. This manuscript further explore which ERα variant was important for FMD. The authors reported a novel ligand-independent pathway leading to flow mediated dilation attributed to the membrane-bound ERα. The authors used mouse model with fully deficient ERα, as well as mouse with specific inactivation of either membrane-bound or nuclear ERα and showed that only disruption of membrane-bound ERα greatly inhibit flow mediated dilation. The author further revealed the mechanism involving shear stress-induced attenuation of ROS levels which increase eNOS activity. Therefore, membrane-bound ERα could be a potential target therapy to reduce oxidative damage in endothelium of resistance arteries.

Strengths

This study presented for the first time that flow mediated dilation produced by estrogen receptor alpha is attributed to its membrane receptor via ligand-independent pathway.

Weaknesses

The mechanism of membrane receptor, ligand independent pathway of estrogen receptor alpha that the authors reported lacks novelty as it was already previously described.

*Reviewer #2:*

Multiple animal models have been used to test the contribution of membrane estrogen receptors on the vascular dilation induced by flow (flow-mediated dilation, FMD).

The authors propose that the presence of membrane estrogen receptors optimizes flow-mediated dilation. However, the conclusion that ER promotes NO production and inhibits oxidative stress is not fully supported by the data since basal ROS production is not altered in the models without functional ER, and an antioxidant treatment normalizes the dilatory response, and effect that can be independent of a direct NOS activation. The data show that in the absence of functional membrane receptors, flow-mediated dilation is reduced but that it is restored by a treatment with an antioxidant. Thus, these receptors seem not to be necessary for FMD. The involvement of these receptors in FMD remains therefore questionable.

*Reviewer #3:*

The Authors report that blood flow in mouse resistance arteries activates membrane-associated ERα, which reduces oxidative stress (O_2_-.), resulting in enhanced NO bioavailability. The absence of membrane-associated ERα may lead to the production of O_2_-. which attenuates NO-dependent dilation as shown by a rise in H_2_O_2_ generation in the perfused kidney.

Strengths

(1) Use of cutting-edge animal models to explore the role of ERα in the vascular endothelium.

(2) Robust experimental procedures including kidney perfusion.

(3) Comprehensive set of experiments including appropriate controls.

(4) Disclosure of a new ligand-independent vascular ERα effect.

Weaknesses

(1) Ignoring a possible role of GPER and signaling pathways thereof in the study endpoints is a major flaw in the experimental design.

(2) The regulation of flow-mediated dilation by estrogen has been widely investigated in previous studies.

(3) The gender claim in the title is relatively weak as it is based on just one set of experiments (Figure 1; lines 137-138).

Although the study objective was stated concisely, the Authors generated findings of potential impact in the field that deserve further investigation. A cross-talk between blood flow, NO pathway and membrane-associated ERα appears to emerge from the present work and represents a conceptual advance. However, the role of GPER in this setting deserves to be assessed as well.

*Reviewer #4:*

Using a variety of genetically-manipulated mouse models, the authors study how membrane associated estrogen receptor alpha (ERα) impacts flow-mediated dilation (FMD). Their collective findings indicate that non-nuclear actions of plasma membrane-associated ERα in endothelial cells support FMD in both sexes via mechanisms that are independent of estrogens. The process likely involves the promotion of NO production and blunting of impact of reactive oxygen species (ROS).

Strengths include the use of multiple genetic manipulations in mice, which allow testing of ERα function as a transcription factor and as a modulator of extra-nuclear signaling initiated by a subpopulation of plasma membrane-associated ERα. The specific requirement for ERα in endothelial cells is also evaluated. FMD is primarily tested in isolated mesenteric resistance arteries, but some key findings are confirmed in uterine arteries. The complementary use of an isolated perfused kidney model is also a strength.

One weakness is the evaluation of levels of ERα expression by quantifying transcript abundance in whole arteries when ERα protein abundance in endothelial cells is of prime importance. In addition, although alterations in NO and ROS are implicated, no insights are gained into how these are impacted by ERα presumably independent of estrogens. Despite the lack of more mechanistic interrogation, the overall observation of an important role for non-nuclear function of plasma membrane-associated ERα in endothelial cells in FMD is important to the field.

[Editors' note: further revisions were suggested prior to acceptance, as described below.]

Thank you for submitting your revised manuscript "Membrane estrogen receptor alpha (ERα) optimizes flow-mediated dilation in both sexes, in a ligand-independent manner" for consideration by *eLife*. Your revised article has been evaluated by a Reviewing Editor and a Senior Editor. the Reviewing Editor has drafted this letter to help you prepare a revised submission.

Essential revisions:

1) The wording "optimization" in the title may be "optimized". The reviewer thinks that e.g. an experimental procedure but not a biological process such as FMD may be optimized. ER participates in or supports FMD.

2) The p values and other statistical parameters could be more appropriately moved from figures to figure legends, especially when asterisks are present.

3) qPCR experiments: "relative expression" may refer to either an housekeeping gene or a reference control sample. Please indicate the first option in figure legends.

4) Figure 5: Western blots should be shown for phospho-eNOS and total eNOS, and for phospho-Akt and total Akt.

5) Line 63-65: The authors state that flow-mediated dilatation depends mainly on NO, citing studies looking at the forearm circulation in humans. This should be stated/clarified. However, in mice (the organism used for experiments in this study) flow-dependent vasodilatation in mesenteric resistance arteries (the vascular bed investigated in this study) is mediated by both, NO and EDH (PMID 16055522; 11282893). Brandes and associates also found that EDH is the main mediator of endothelium-dependent relaxation in murine resistance arteries (PMID 10944233). Moreover, the EDH-shear-dependent response is partly sensitive to non-selective COX-inhibition (PMID 16055522). The authors should discuss these important differences between humans and mice and also should mention the limitation of their study that experiments were conducted in the absence of COX inhibition, and that one cannot fully exclude the involvement of COX-dependent / endothelial-derived prostanoid effects in the effects observed.

6) Lines 112-115: With regard to estrogen effects in men the authors should discuss intracrine, aromatase-mediated production of estradiol which is converted locally in the vasculature from testosterone (PMID 11248122) by which estrogen partly provides protection in male mice from atherosclerosis.

7) Lines 145-147: The authors suddenly introduce compounds targeting GPER, without having made any reference to this receptor in the introduction. It would be helpful if the authors could add a little section to the introduction discussing this receptor as well, also citing a good overview article, such as PMID 21844907

8) Line 144, Line 465-466, : The authors state "the ERα-ERb antagonist and GPER agonist fulvestrant (ICI-182780)." This is partly correct. Fulvestrant is a SERD (selective estrogen receptor downregulator or selective estrogen receptor degrader), which down-regulates/degrades the receptors it targets. This should be corrected.

9) Line 262: findings for the evaluation of changes of gene expression with C451A-ERα are not shown in Figure 7.

10) Lines 335-336: the statement is misleading because chronic effects of E2 (recognizing that the term "chronic" is vague) can be mediated by non-nuclear actions of ERα.

11) Lines 402-403: there is no basis for stating that the decrease in FMD in mice lacking membrane ERα could reflect a feature of premature aging of the endothelium.

12) Figure 9 schematic: there is no evidence that Gai mediates the role of membrane-associated ERα in FMD.

13) P. 21: The reference for G36 provided in the table is not quite correct, the one cited was published to 2010, at a time when G36 had not yet been published. It is suggested to list PMID 27803283 as reference instead.

[Editors' note: further revisions were suggested prior to acceptance, as described below.]

Thank you for resubmitting your work entitled "Membrane estrogen receptor alpha (ERα) participates in flow-mediated dilation in a ligand-independent manner" for further consideration by *eLife*. Your revised article has been evaluated by a Reviewing Editor and a Senior Editor.

The manuscript has been improved but there are some remaining issues that need to be addressed, as outlined below:

1) Previous comment "Lines 145-147: The authors suddenly introduce compounds targeting GPER, without having made any reference to this receptor in the introduction. It would be helpful if the authors could add a little section to the introduction discussing this receptor as well, also citing a good overview article, such as PMID 21844907"

Authors Response: We have added a paragraph in the introduction stating the role of GPER in endothelium-dependent dilation (lines 87-90):

"E2-dependent vasodilation has also been shown to involve the G-protein-coupled estrogen receptor (GPER) in both human and animal arteries [24]. In the rat, GPER activation reduces uterine vascular tone during pregnancy through activation of endothelium-dependent NO production [25]."

It appears that there was a misunderstanding with regard to the previous comment. It was not requested to describe solely the role of GPER in endothelial cell function and NO release. Rather, the authors were expected to expand this section, briefly describing the nature of GPER (namely that it is a 7-transmembrane GPCR located at the endoplasmic reticulum), its natural and synthetic ligands, and its functions (release of NO is just one). Also, the authors should mention that there is both, ligand-dependent and ligand-independent activation (PMID 27803283) of GPER, which should also be discussed in the Discussion section.

2) The "Key Resources Table" in the Methods section includes a number of references:

Hypertension 2007;50:248-54

Hypertension. 2007;50:248-54

EMBO Mol Med. 2014;6:1328-46

Pharmacology 2010: 86, 58- 64

Sci Signal. 2016;9:ra105

J Vasc Res. 2009;46:2 53-64

Am J Physiol 2018;315:H1019-H1026

Cytokine 2009;46,166-70

These references should be added in full to the References section

3) The whole manuscript should be checked for grammar and typos, including "contribute to relaxation of the mesenteric resistance arteries in both male and female through, at least in part, a PI3K-Akt-eNOS pathway [28]." (lines 98-99, page 5) and corrections should be made as needed.

---

## [Author Response]

Essential revisions:1) The ligand-independent activation of membrane ERα is a novel finding. However, simply ignoring a potential role of GPER in such ligand-independent regulation of FMD is a major flaw. The authors should carry out some straight-forward studies to look at the effects of pharmacological inhibition using GPER antagonist, G-36, which turns out to not only inhibit GPER but also via inhibition of constitutive GPER expression, to reduce abundance (and activity) of Nox1. The authors are advised to perform in vitro studies of vessels from WT mice, evaluating dilation in response to flow +/- GPER antagonist, as long as a parallel control study is done evaluating dilation in response to GPER agonist +/- GPER antagonist.

We agree with your comment. Ligand-independent activation of membrane ERα has been previously demonstrated. However, the present finding shows, for the first time its involvement in the endothelium response to flow in resistance arteries.

We performed the experiments requested and added the data to figure 1 (panels I and J) and to the corresponding paragraph in the manuscript (Results section on figure 1 and discussion). We found that G36 incubation did not affect flow-mediated dilation in arteries from WT mice suggesting that GPER may not have a role in this vascular response although it is now clear that GPER has a full place in cardiovascular homeostasis and pathophysiology.

As G36 has an IC50 for G1 and E2 (estradiol) which are very close (165 nM and 112 nM, respectively: Dennis et al., J Steroid Biochem Mol Biol. 2011 Nov;127(3-5):358-66), we tested the effect of G36 on both compounds and observed that G36 inhibited both dilation indued by G1 and E2 (panel J, figure 1).

2) In the Introduction section, the authors are encouraged to cite original works related to non-nuclear signaling by the subpopulation of ER associated with the plasma membrane in endothelium.

We have extended this part of the discussion (Introduction, lines 74 to 93).

3) In the Introduction section, the background information about cardiovascular disease risk in men versus women and the impact of estrogens on risk is not as clear-cut as the authors imply. A more balanced presentation of the clinical evidence is warranted.

We have added more studies describing the protective effect of estrogens with a focus on FMD. This is mainly in the introduction between line 92 and line 111.

4) In the Introduction, a more detailed account of the published literature in the field including studies in humans should be provided The Authors may consider citing some studies from among PMID 19126786; 7642876; 20185791; 26471832; 9176294; 17367797; 28645860; 26734763; 9791075; 16131583; 18319309; 19086257; 9930647.

We have included most of these articles in the introduction and/or in the discussion.

Several articles are from our group and show that flow-mediated outward remodeling involves E2-dependent activation of nuclear function (AF2) of ERα. This remodeling occurs in response to a chronic increase in flow and affects all the layers of the arterial wall. By contrast with FMD which is an acute response involving non-nuclear ERα. We have added a paragraph on this difference to avoid confusion (lines 70-74 and in the discussion: lines 337-349).

5) In the Results section, Page 6, line 106 incorrectly refers the reader to Figure 1B.

We have corrected this error (lines 135-37).

6) In the Results section. Page 6, line 117: fulvestrant is an ERα and ERβ antagonist (pKi = 9 for both) as well as a GPER full agonist (pKi = 7).

We have completed the corresponding text as follows:

“Furthermore, the ERα-ERb antagonist and GPER agonist fulvestrant (ICI-182780) did not alter the FMD” Now in line 144.

Same correction in the Material and Methods section.

7) In the Results section, Figures 2B, 2F and 2J: ERα protein abundance in endothelial cells should be evaluated, even if only feasible from vasculature that would yield more endothelial cells than mesenteric resistance arteries.

We have performed these experiments in endothelial cells isolated from the aorta as too few cells can be obtained from the mesenteric arteries. This new data is shown in Figure 3 B, F and J.

Besides markers of endothelial cells (Tie2 or *Tek*) and of smooth muscle cells (*Cnn1*) were quantified in order to control for the quality of the endothelial cells isolation.

*Tek* and *Cnn1* expression levels are in Figure 3, supplement figure 1.

8) In the Results section, Figure 5B: the flow-pressure relationship was shifted left, or up, in C451A-ERα mice versus WT according to the group labeling, and not to the right.

We have corrected this text. Thank you for having noticed this error.

9) In the Results section Page 17, line 281: the use of the term autoregulation here may be confusing, and more specific interpretation would be helpful (and better placed in the Discussion section).

We have modified the sentence as follows:

“Thus, these results suggest that FMD reduction due to the absence of membrane-ERα also affects the capacity of the renal vasculature to produce NO and ATP”

Now in Lines 254 – 257.

10) In the Results section, how does flow increase eNOS Ser1177 phosphorylation and alter H_2_O_2_ via ERα mechanistically ? Are flow-related changes in Akt phosphorylation in endothelium altered in the absence of ERα?

To respond to these questions, we have added several sets of new data.

We have added new data showing the level of phosphorylation of Akt in AF2^0^ERα, C451A-ERα and R264-ERα mice (and the corresponding WT groups) (Figure 5, panels C, F and I + Figure 5 source data 2 showing all the blots for eNOS, Ph-eNOS, Akt and Ph-Akt).

In addition, we have added new data showing that mito-tempo which reduces ROS production by the mitochondria improved FMD in C451A-ERα mice (figure 7). In figure 7 we also added new data showing that the combination of superoxide dismutase and catalase also improved FMD in arteries from C451A-ERα mice (no effect in WT) and that catalase alone (elimination of H_2_O_2_ into H2O) reduced FMD in C451A-ERα mice without affecting FMD in WT.

Thus, we may state that ERα reduces mitochondria functioning and its production of ROS and H_2_O_2_. Indeed, several studies have shown that flow (shear stress) induces a more quiescent and less oxidative phenotype in endothelial cells (i.e.: Doddaballapur et al. Arterioscler Thromb Vasc Biol 2015**,** 35, (1), 137-45). This reduction in mitochondrial ROS production would thus allow NO to be more efficient (less ROS scavenging NO) and allow a more efficient eNOS activation. Of course, this later issue remains to be further investigated. Nevertheless, an excessive ROS production has been shown to reduce eNOS phosphorylation through increased phosphatase activation (Ding et al. Front Physiol 2020, 11, 566410).

In addition, the effect of catalase on FMD in C451A-ERα mice agrees with previous studies by D. Gutterman’s group showing that H_2_O_2_ produced by the mitochondria can dilate coronary arteries in response to flow (FMD) in arteries from patients with coronary artery disease. Although this allows keeping some FMD, the dilation remains low (as in C451A-ERα mice in the present study) and in the long term, H_2_O_2_ remains deleterious.

We have also modified the scheme shown in figure 9 to include these new data.

11) In the Results section, Do non-nuclear actions of plasma membrane-associated ERα influence endothelial cell production of prostaglandins in the setting of FMD?

We have added new data showing that the cyclooxygenase inhibitor indomethacin does not further reduce FMD in the 4 groups of mice studied: ERα-/-, AF2^0^ERα, C451A-ERα and R264-ERα mice (and the corresponding WT groups) (new data added to Figure 4).

In addition, we observed no change in COX1, COX2 and prostacyclin expression level in mesenteric arteries isolated from C451A-ERα mice (Figure 7—figure supplement 2: J, K and L).

EDHF is the third major agent produced by the endothelium in response to flow and it is also involved in the activation of endothelium-dependent dilation by estrogens (reference 31 and 32). Thus, we have also added to the manuscript new data obtained with EETs blocker MSPPOH as EETs are major members of the EDHF family. EETs have been shown to mediate, at least in part, the protective effect of E2 on FMD in hypertensive or old rats (reference 31 and 32, introduction, lines 103-109).

This data (indomethacin and MSPPOH) is discussed lines 360-365.

In Figure 4 we show only L-NNA and indomethacin to avoid overloading the figure.

Figure 4 supplement figure 1 shows the 3 blockers, L-NNA, indomethacin and MSPPOH.

12) In the Results section, do non-nuclear actions of plasma membrane-associated ERα influence mechanosensitive ion channel localization or function in endothelial cells?

Based on the experiments described above, it is most likely that non-nuclear actions of plasma membrane associated ERα involve a reduction in oxidative stress and subsequently a better action of eNOS.

Nevertheless, it is important to decipher the link between flow (shear stress) activation of the extracellular matrix and cell surface and the plasma membrane associated ERα.

As flow has been recently shown to activate Piezo1-dependent release of ATP through pannexin hemi-channels followed by ATP-dependent activation of purinergic receptors which induce NO production by endothelial cells (Wang et al. J Clin Invest 2015, 125, (8), 3077-86; Wang et al., J Clin Invest 2016, 126, (12), 4527-4536), we investigated this pathway in C451A-ERα mice:

First, we found no change in the expression level of Piezo1 and Piezo2 channels in mesenteric arteries isolated from C451A-ERα mice. Similarly, no change in the level of polycystin1 and 2, TRPV4 or integrin α and β was found and no change in purinergic receptors level was found (Figure 7—figure supplement 1: panel G to N, Q, R and S).

Second, we have also performed additional experiments and added the new data to the manuscript showing that the acute response to ATP and to the piezo1 agonist YODA-1 was not affected by the absence of membrane ERα in C451A-ERa mice.

Similarly, the mechanosensitive channel blocker GsMTx4 similarly affected FMD in C451A-ERα and WT mice suggesting that the defect in FMD is probably located down-stream flow sensing. New data added to the discussion, lines 376-381 and shown in Figure 7—figure supplement 3, A, B and C.

As discussed above (point 11), we have also included data obtained with the EETs synthesis blocker MSPPOH. Indeed, as stated above, we used indomethacin to assess the role of prostaglandins in FMD in the mouse models used in the present work. We also used MSPPOH as EETs are produced by the endothelium and activate transient receptor potential (TRP) channel (Campbell WB, Fleming I. Epoxyeicosatrienoic acids and endothelium-dependent responses. Pflugers Arch. 2010 May;459(6):881-95). Nevertheless, this compound did not significantly change FMD after L-NNA and indomethacin blockade suggesting that this pathway and the related channels may not be involved (Figure 4 -, figure supplement 1, shown above in response to point 11).

13) In the Results section, why was the antioxidant TEMPOL specifically selected? Would the Authors expect a similar outcome following treatment with other antioxidants such as e.g. quercetin or dimethyl fumarate?

We used TEMPOL as we have already used it in the past with success. In addition, it is easy to use in the drinking water and mice keep drinking normally.

Nevertheless, we have added new data obtained with mice treated with vitamin C and vitamin E (4 weeks of treatment). This treatment has been shown efficient (Favre J et al. Coronary endothelial dysfunction after cardiomyocyte-specific mineralocorticoid receptor overexpression Am J Physiol Heart Circ Physiol. 2011 Jun;300(6):H2035-43) in reducing ROS.

This new set of experiments shows a similar result than with Tempol with a restoration of FMD in treated C451A-ERα mice treated with the antioxidant cocktail vitamin C and vitamin E. This new data is presented in figure 8 (G to L).

Importantly, we obtained a similar effect with 2 different unrelated antioxidant treatments.

14) In the Results section, does Tempol or PEG-catalase acutely restore FMD ex vivo?

We have added new data showing that the combination of superoxide dismutase and catalase restores FMD in C451A-ERa mice. By contrast, catalase which reduces H_2_O_2_ level improved FMD in C451A-ERa mice. This result supports the assumption that H_2_O_2_ may be responsible for the remaining FMD in C451A-ERa mice. This was not observed in WT mice. In addition, the effect of superoxide dismutase + catalase which reduces more globally the production of reactive oxygen species improved FMD in C451A-ERa mice. A similar result was obtained with mito-tempo which reduces mitochondrial ROS. This is discussed in more details above (point 10). New data shown in figure 7.

15) In the Results section, Line 312: Chronic for a 2-week treatment does not sound very appropriate.

We have removed the word chronic and left the duration of the treatment (2 weeks).

Of note, the new antioxidant treatment (vitamin E/vitamin C) added to the revised manuscript was supplied for a longer duration (4 weeks). This is now in figure 8.

16) In the Results section, flow rate may be changed to shear stress in Dyn/cm2: this would help comparing between published works in the topic.

We have added the corresponding shear stress in the Material and Methods and in the figure legends.

17) The Discussion section should be shortened, providing more succinct focused discussion of the interpretation of the findings, their implications, and possible explanations to fill the new knowledge gaps that result from the work.

We shortened the initial discussion. Nevertheless, we also added some text to respond to the comments raised above. Altogether, the discussion remains shorter and hopefully better focused on the results of the present work.

18) In the Conclusion section, Figure 7: the schematic is only in part helpful because how non-nuclear actions of membrane-associated ERα in endothelial cells govern eNOS and ROS in response to flow is not addressed.

We have modified this scheme (now Figure 9) and hope that it now better supports the conclusion of the discussion.

Reviewer #1:[…] The mechanism of membrane receptor, ligand independent pathway of estrogen receptor alpha that the authors reported lacks novelty as it was already previously described.

We agree that ligand independent involvement of estrogen receptor alpha has been described in other fields, mainly in the field of breast cancer. Nevertheless, its involvement in the acute response to flow of resistance arteries is new and potentially of interest in the field of cardiovascular diseases.

Reviewer #2:Multiple animal models have been used to test the contribution of membrane estrogen receptors on the vascular dilation induced by flow (flow-mediated dilation, FMD).The authors propose that the presence of membrane estrogen receptors optimizes flow-mediated dilation. However, the conclusion that ER promotes NO production and inhibits oxidative stress is not fully supported by the data since basal ROS production is not altered in the models without functional ER, and an antioxidant treatment normalizes the dilatory response, and effect that can be independent of a direct NOS activation. The data show that in the absence of functional membrane receptors, flow-mediated dilation is reduced but that it is restored by a treatment with an antioxidant. Thus, these receptors seem not to be necessary for FMD. The involvement of these receptors in FMD remains therefore questionable.

We agree that the link between membrane ERα and eNOS activation by flow remains an open question. To address this question, we have added new data to the manuscript which should help better defining the relation between membrane ERα and the balance between eNOS activity and ROS.

First, we have added data showing that ROS inhibition is also active acutely and restored FMD in vitro in arteries isolated from mice lacking membrane-ERα (C451A-ERα mice) in agreement with the in vivo data obtained previously (initial version of the manuscript) showing that a 2-weeks long treatment with antioxidant TEMPOL restored FMD in C451A-ERα mice (we have also included another treatment with vitamins C and E, 4 weeks). This is also in agreement with our measurement of H_2_O_2_ in the perfusate of isolated kidneys from C451A-ERα mice. Thus, membrane-ERα is necessary to reduce ROS production, and this allows a better FMD. Therefore, FMD is reduced in the absence of membrane-ERα (no more break on ROS production) and ROS reduction restored FMD. Of course, this could exclude a direct involvement of membrane-ERα in flow-sensing and signal transduction to eNOS as you point out in your comment. To further address this question, we used blockers of mechanosensitive channels (similar effect on C451A-ERα and WT mice) and we tested Yoda1 and ATP-dependent dilation in C451A-ERα and WT mice (no difference).

Thus, it seems that flow activates on one hand the NO pathway and on the other hand flow reduces ROS production through activation of the membrane located ERα. This effect on ROS could involve a reduction in mitochondrial activity as we also found that the inhibition of mitochondrial ROS production with Mito-Tempo restored FMD in C451A-ERα mice (new data added to the manuscript). We have extended the discussion on this mechanism. Of course, the pathway linking flow to membrane-ERα and to the mitochondria remains to be further investigated.

Altogether, these data show that membrane-ERα is involved in the acute response to flow of resistance arteries through a reduction in ROS production.

We have changed the conclusion of the abstract as follows:

“Thus, endothelial membrane ERα promotes NO production through inhibition of oxidative stress and thereby helps to optimize FMD in a ligand-independent manner”

Reviewer #3:[…] (1) Ignoring a possible role of GPER and signaling pathways thereof in the study endpoints is a major flaw in the experimental design.

We have tested the effect of G36 as suggested. Although G36 did not affect FMD in the present study, GPER is certainly an important player in the control of vascular tone in different conditions. Although GPER does not seem to have a role in FMD in the present work, its role in the pathophysiology of the vascular tree is now well recognized.

(2) The regulation of flow-mediated dilation by estrogen has been widely investigated in previous studies.

We agree that estrogen has major role in restoring FMD in many pathological (cardiovascular and metabolic disorders) or physiological (menopause) conditions. We have added more references on this effect of estrogen in the introduction (lines 96-111).

Nevertheless, the present study does not address the effect of estrogen on FMD. On the contrary, we show in the present study that membrane ERα is involved in FMD independently of its ligand. This effect was observed in healthy young mice, both in females and in males. Importantly, this effect is reminiscent of the first case of ERα gene deficiency in a young man: “The first disruptive mutation in the ERα gene, reported in 1994 in a man who was only 30 years old, was found to be associated with a selective and total absence of FMD. This single yet major clinical observation suggests that ERα-dependent signal transduction could play a role in FMD in males”.

Smith et al. N Engl J Med 1994, 331, (16), 1056-61.

Sudhir et al. Lancet 1997, 349, (9059), 1146-7.

3) The gender claim in the title is relatively weak as it is based on just one set of experiments (Figure 1; lines 137-138)

We agree that most of the experiments were conducted on male mice. Nevertheless, the main effect of the absence of membrane-ERα is a reduction of the amplitude of FMD in both male and female mice. This was observed in mesenteric arteries from intact and ovariectomized mice as well as in the uterine arteries of female mice. This later data was shown in the supplement files, and we moved it to figure 1. Nevertheless, we have removed “male and female mice” from the title.

Although the study objective was stated concisely, the Authors generated findings of potential impact in the field that deserve further investigation. A cross-talk between blood flow, NO pathway and membrane-associated ERα appears to emerge from the present work and represents a conceptual advance. However, the role of GPER in this setting deserves to be assessed as well.

We agree with your comment and have performed the requested experiments. We found that G36 did not affect FMD in the mouse mesenteric artery. We have added the data to figure 1.

Reviewer #4:[…] One weakness is the evaluation of levels of ERα expression by quantifying transcript abundance in whole arteries when ERα protein abundance in endothelial cells is of prime importance. In addition, although alterations in NO and ROS are implicated, no insights are gained into how these are impacted by ERα presumably independent of estrogens. Despite the lack of more mechanistic interrogation, the overall observation of an important role for non-nuclear function of plasma membrane-associated ERα in endothelial cells in FMD is important to the field.

We have isolated endothelial cells from the aorta to quantify ERα gene expression (data in figure 3). We used the aorta as the quantity of cells that would be obtained from mesenteric resistance arteries would be too small and the isolation techniques used for the aorta does not apply to small resistance arteries. Although more cells are obtained from the aorta, the quantity of cells and consequently the quantity of total tissue and proteins obtained remains very limited. Analysis of the amount of protein in endothelial cells isolated from the mouse vessels is not feasible due to the insufficient amount of biological material combined to the low efficiency of ERα antibody in mice. However, we have adapted the protocol described by Iruela Arispe's team to analyze the expression of ERα in endothelial cells isolated from aorta by Q-PCR (Briot et al., Repression of *Sox9* by Jag1 is continuously required to suppress the default chondrogenic fate of vascular smooth muscle cells. Dev Cell. 2014;31(6):707-721).

The results obtained demonstrated that there is no difference in the expression of ERα in the endothelial cells, whatever the genotype of the mice. Besides markers of endothelial cells (Tie2) and of smooth muscle cells (Cnn1) were quantified in order to control for the quality of the endothelial cells isolation. ERα is shown in figure 3. Tie2 (*Tek*) and *Cnn1* are shown in the supplemental figures (Figure 3—figure supplement 1).

[Editors' note: further revisions were suggested prior to acceptance, as described below.]

Essential revisions:1) The wording "optimization" in the title may be "optimized". The reviewer thinks that e.g. an experimental procedure but not a biological process such as FMD may be optimized. ER participates in or supports FMD.

As recommended, the title is now:

“Membrane estrogen receptor alpha (ERα) participates in flow-mediated dilation in a ligand-independent manner”.

We have also modified the last sentence of the abstract.

2) The p values and other statistical parameters could be more appropriately moved from figures to figure legends, especially when asterisks are present.

As recommended, p values are now given in the figure legends.

Data and full statistical analysis are also shown in the source data files.

3) qPCR experiments: "relative expression" may refer to either an housekeeping gene or a reference control sample. Please indicate the first option in figure legends.

As recommended, the text of the figure legend (figure 3) is now:

*“Esr1* expression level in aortic endothelial cells (expression relative to the housekeeping genes *Gapdh, Hprt and Gusb*)”

4) Figure 5: Western blots should be shown for phospho-eNOS and total eNOS, and for phospho-Akt and total Akt.

As recommended, Western-blots (selected bands) are now shown in figure 5.

Western blots for all the mice used in the study are shown in the Figure 5 source data 2.

5) Line 63-65: The authors state that flow-mediated dilatation depends mainly on NO, citing studies looking at the forearm circulation in humans. This should be stated/clarified. However, in mice (the organism used for experiments in this study) flow-dependent vasodilatation in mesenteric resistance arteries (the vascular bed investigated in this study) is mediated by both, NO and EDH (PMID 16055522; 11282893). Brandes and associates also found that EDH is the main mediator of endothelium-dependent relaxation in murine resistance arteries (PMID 10944233). Moreover, the EDH-shear-dependent response is partly sensitive to non-selective COX-inhibition (PMID 16055522). The authors should discuss these important differences between humans and mice and also should mention the limitation of their study that experiments were conducted in the absence of COX inhibition, and that one cannot fully exclude the involvement of COX-dependent / endothelial-derived prostanoid effects in the effects observed.

We have modified the text as recommended.

More precisely:

a. Line 63-65: it is now stated that this sentence refers to human studies.

The sentence is now: “FMD measured in the human forearm depends mainly on the acute production of NO…” Line 64 now.

b. In mice FMD in mesenteric resistance arteries is mediated by both, NO and EDH:

We have added a paragraph in the discussion stating the limitations of the study as suggested by your comment. Line 365 to 374. As regard to COX-derivatives, we have added to figure 4 (in response to the previous comments) data obtained with indomethacin. Indeed, the addition of indomethacin to LNNA did not further reduced FMD and acetylcholine-mediated dilation. Similarly, the addition of MSPPOH which block EETs production, did not further reduced FMD and acetylcholine-mediated dilation (supplement figure to Figure 4). Although EETs are major component of the EDHF family, this result suggests that the remaining FMD in the presence of L-NNA, indomethacin and MSPPOH is due to other EDHFs.

6) Lines 112-115: With regard to estrogen effects in men the authors should discuss intracrine, aromatase-mediated production of estradiol which is converted locally in the vasculature from testosterone (PMID 11248122) by which estrogen partly provides protection in male mice from atherosclerosis.

We have added the following sentence in the introduction (now: lines 118-120)

“Of note, testosterone has been shown to reduce early atherogenesis in male mice through its conversion to estrogen by aromatase which is expressed in the arterial wall [42]”.

7) Lines 145-147: The authors suddenly introduce compounds targeting GPER, without having made any reference to this receptor in the introduction. It would be helpful if the authors could add a little section to the introduction discussing this receptor as well, also citing a good overview article, such as PMID 21844907

We have added a paragraph in the introduction stating the role of GPER in endothelium-dependent dilation (lines 87-90):

“E2-dependent vasodilation has also been shown to involve the G-protein-coupled estrogen receptor (GPER) in both human and animal arteries [24]. In the rat, GPER activation reduces uterine vascular tone during pregnancy through activation of endothelium-dependent NO production [25].”

8) Line 144, Line 465-466, : The authors state "the ERα-ERb antagonist and GPER agonist fulvestrant (ICI-182780)." This is partly correct. Fulvestrant is a SERD (selective estrogen receptor downregulator or selective estrogen receptor degrader), which down-regulates/degrades the receptors it targets. This should be corrected.

We have corrected the text: “the estrogen receptor downregulator and GPER agonist fulvestrant (ICI-182780)”. Now: line 149 and line 463

9) Line 262: findings for the evaluation of changes of gene expression with C451A-ERα are not shown in Figure 7.

These findings are shown in the supplement figures attached to Figure 7. This is written in the text as requested in the instruction to the authors: “Figure 7, figure supplement figures 1 and 2” These two supplemental figures were attached to the manuscript (merged files) after the figures (page 68 and 69).

10) Lines 335-336: the statement is misleading because chronic effects of E2 (recognizing that the term "chronic" is vague) can be mediated by non-nuclear actions of ERα.

We removed the sentence “The nuclear functions of ERα are mainly involved in the chronic effects of E2 and ERα activation such as atheroma prevention [7]”

Indeed, you are right, it is a misleading shortcut. In addition, the transition to flow-mediated remodeling is better now.

11) Lines 402-403: there is no basis for stating that the decrease in FMD in mice lacking membrane ERα could reflect a feature of premature aging of the endothelium.

We removed the paragraph. Aging is certainly more complex than a single reduction in membrane-ERα signaling even though it might contribute. A more in-depth investigation is needed before stating this.

12) Figure 9 schematic: there is no evidence that Gai mediates the role of membrane-associated ERα in FMD.

We removed Gai from the scheme.

13) P. 21: The reference for G36 provided in the table is not quite correct, the one cited was published to 2010, at a time when G36 had not yet been published. It is suggested to list PMID 27803283 as reference instead.

We apology for the error. We added the right reference to the table.

[Editors' note: further revisions were suggested prior to acceptance, as described below.]

The manuscript has been improved but there are some remaining issues that need to be addressed, as outlined below:1) Previous comment "Lines 145-147: The authors suddenly introduce compounds targeting GPER, without having made any reference to this receptor in the introduction. It would be helpful if the authors could add a little section to the introduction discussing this receptor as well, also citing a good overview article, such as PMID 21844907".Authors Response: We have added a paragraph in the introduction stating the role of GPER in endothelium-dependent dilation (lines 87-90):"E2-dependent vasodilation has also been shown to involve the G-protein-coupled estrogen receptor (GPER) in both human and animal arteries [24]. In the rat, GPER activation reduces uterine vascular tone during pregnancy through activation of endothelium-dependent NO production [25]."It appears that there was a misunderstanding with regard to the previous comment. It was not requested to describe solely the role of GPER in endothelial cell function and NO release. Rather, the authors were expected to expand this section, briefly describing the nature of GPER (namely that it is a 7-transmembrane GPCR located at the endoplasmic reticulum), its natural and synthetic ligands, and its functions (release of NO is just one). Also, the authors should mention that there is both, ligand-dependent and ligand-independent activation (PMID 27803283) of GPER, which should also be discussed in the Discussion section.

We apologize for misunderstanding your previous comment.

We have extended the paragraph on GPER in the introduction (lines 93-104):

“The 7-transmembrane G-protein-coupled estrogen receptor (GPER, formerly known as GPR30) is another receptor located not only at the plasma membrane but also on the membrane of the endoplasmic reticulum that can be activated by E2. […] Thus, both ERα and GPER could contribute to the rapid actions of E2, although their respective roles according to vessel type, species and pathophysiological context remain to be established.”

We have also extended the paragraph on GPER in the discussion including the various physiological functions involving GPER and the various diseases which could benefit from a targeting of GPER (lines 357-366):

“Another membrane receptor for E2 located at the plasma membrane is GPER [27]. Both ligand-dependent and ligand-independent activation of GPER have been reported [61]. […] However, a possible role of unliganded GPER activation cannot be excluded in case of a crosstalk between membrane-dependent Erα and GPER activation.”

2) The "Key Resources Table" in the Methods section includes a number of references:Hypertension 2007;50:248-54Hypertension. 2007;50:248-54EMBO Mol Med. 2014;6:1328-46Pharmacology 2010: 86, 58- 64Sci Signal. 2016;9:ra105J Vasc Res. 2009;46:2 53-64Am J Physiol 2018;315:H1019-H1026Cytokine 2009;46,166-70These references should be added in full to the References section

References are now in full in the References Section.

3) The whole manuscript should be checked for grammar and typos, including "contribute to relaxation of the mesenteric resistance arteries in both male and female through, at least in part, a PI3K-Akt-eNOS pathway [28]." (lines 98-99, page 5) and corrections should be made as needed.

We have rechecked the manuscript.